# Reconstruction of 3D precipitation measurements from FY-3G MWRI-RM imaging and sounding channels

Yunfan Yang[1,2], Wei Han[3,4], Haofei Sun[1,2], Jun Li[5], Jiapeng Yan[4], and Zhiqiu Gao[1,2]

[1]State Key Laboratory of Atmospheric Boundary Layer Physics and Atmospheric Chemistry (LAPC), Institute of Atmospheric Physics, Chinese Academy of Sciences, Beijing, China
[2]University of Chinese Academy of Sciences, Beijing, China
[3]CMA Earth System Modeling and Prediction Centre (CEMC), China Meteorological Administration, Beijing, China
[4]State Key Laboratory of Severe Weather (LaSW), Chinese Academy of Meteorological Sciences, China Meteorological Administration, Beijing, China
[5]National Satellite Meteorological Center, China Meteorological Administration, Beijing, 100081, China

**Correspondence:** Wei Han (hanwei@cma.gov.cn)

**Abstract.** FengYun 3G satellite (FY-3G), China's first precipitation measurement satellite, was launched on April 17, 2023. FY-3G carries an advanced multi-channel microwave radiance imager-rainfall measurement (MWRI-RM) system, which, compared to the previous GPM/GMI, includes more sounding channels. Additionally, a Ka/Ku-band dual-frequency precipitation measurement radar (PMR) onboard FY-3G provides 3D observations of severe precipitation systems. Due to the high cost and hardware limitations of precipitation radars, most precipitation-affected satellite observations rely on passive data. Deep learning methods have become effective tools to bridge these two types of observations. In this study, we proposed a deep convolutional neural network (CNN) to reconstruct PMR-Ku reflectivity profiles based on MWRI-RM multi-channel radiances across different precipitation scenarios and analyzed the effects of dual oxygen absorption sounding channels and polarization differences (PD) on reconstruction outcomes. Experiments showed that dual oxygen absorption sounding channels improved reflectivity profiles accuracy, especially under land and coastal precipitation scenarios, with RMSEs reduced by 5.43% and 5.47%. Including PD further enhanced reconstruction accuracy and highlighted the critical role of polarization difference in refining radar reflectivity reconstructions. When applied to Typhoon Khanun (2023) and "23·7" BTH extreme rainfall event, the models demonstrated the combined benefits of dual oxygen sounding channels and PD. These enhancements not only improved accuracy but also enabled a more comprehensive three-dimensional representation of precipitation systems.

## 1 Introduction

Precipitation plays a critical role in the Earth's climate system and water cycle, influencing everything from global climate patterns to agricultural productivity and water resources (Huang et al., 2023). Understanding the instantaneous structure and global distribution of precipitation is essential for comprehending changes in the global climate. Satellite observations, combining passive microwave sensors (PMW) and radars, have become vital tools in enhancing our understanding of these processes (Dubovik et al., 2021). They provide comprehensive coverage over diverse terrains, including oceans, deserts, and high plateaus, thus deepening our knowledge of atmospheric dynamics on a global scale (Hou et al., 2014).

The progression of space-based precipitation measurements began with the launch of the Tropical Rainfall Measuring Mission (TRMM) in 1997 (Simpson et al., 1996). TRMM was equipped with a suite of sensors, including the Precipitation Radar (PR) and the TRMM Microwave Imager (TMI), enabling combined passive-active microwave observations and allowing for detailed analyses of the macro- and microphysical structures of precipitating clouds (Kummerow et al., 1998). Building on the legacy of TRMM, the Global Precipitation Measurement (GPM) mission was initiated, with the GPM Core Observatory (GPMCO) launched in 2014 (Draper et al., 2015). Orbiting at an altitude of 407 km with a 65° inclination, GPMCO carrying a Ka/Ku-band dual-frequency precipitation radar (DPR) and a multifrequency microwave imager (GMI). The GMI and DPR also serve as a transfer standard for calibrating precipitation retrieval algorithms by establishing a reference for other satellites in the GPM constellation. The mission's expanded capabilities facilitated a deeper understanding of the three-dimensional structures and global distribution of precipitation (Draper and Newell, 2018).

In response to these developments and the significance of hazardous weather events like typhoons, heavy rains, and severe convective storms in China, the China Meteorological Administration, in collaboration with the China Aerospace Science and Technology Corporation, initiated the development of the Fengyun-3 (FY-3) series of precipitation satellites. On April 16, 2023, at 09:36 Beijing time, China's first dedicated precipitation measurement satellite, FY-3G, was successfully launched. FY-3G is equipped with the passive Microwave Radiance Imager-Rainfall Measurement (MWRI-RM) and the active Precipitation Measurement Radar (PMR), enhancing the detection of atmospheric, cloud, and precipitation structures and advancing the scientific understanding of precipitation and cloud microphysics (Zhang et al., 2023a). Building on the microwave imager (MWRI) technology of the FY-3 satellite series, MWRI-RM integrates microwave sounding and imaging capabilities with 26 channels ranging from 10 to 183 GHz (He et al., 2024). In addition to carrying channels similar to those on previous microwave imagers like TMI and GMI, MWRI-RM includes additional sounding channels around the 50 GHz and 118GHz oxygen absorption bands (Table.1). The dual oxygen absorption sounding channels offer several unique advantages. Firstly, due to their distinct absorption and scattering characteristics (Bauer and Mugnai, 2003; Bauer et al., 2005), the combination of 50 GHz and 118 GHz channels can effectively identify clouds and precipitation (Hu and Weng, 2022). Furthermore, cloud structures can be retrieved at various height levels, enabling three-dimensional observations (Han et al., 2015). Secondly, compared to traditional window channels, the dual oxygen absorption sounding channels are less sensitive to land surface emissivity, making them more suitable for all-surface observing and monitoring of the atmosphere (Prigent et al., 2005). Finally, these channels can also provide rich and valuable information about the atmospheric thermal structure (Carminati et al., 2021). The inclusion of auxiliary sounding channels near the 183 GHz water vapor lines further enhances MWRI-RM's capability on profiling the atmospheric humidity and precipitation (Laviola and Levizzani, 2011; Laviola et al., 2013). A preliminary assessment of MWRI-RM/FY-3G brightness temperatures (BT) using double difference (DD) analysis, based on simultaneous measurements from GMI, suggests that MWRI-RM has the performance comparable to GMI for most channels, particularly the channels at 166 GHz and 183 GHz (He et al., 2024).

Previous studies have shown that there is a strong correlation between active and passive microwave observations. For instance, active observations have been used to validate precipitation profiles retrieved from passive radiance data (Bauer and Mugnai, 2003; Kummerow et al., 1991), and combined active-passive microwave observations have been instrumental in

retrieving comprehensive precipitation products and classifying precipitation types (Grecu et al., 2004, 2016; Das et al., 2022). The complementary data from both methods provide valuable insights into precipitation characteristics. Despite the ability of radar to provide detailed three-dimensional (3D) observations, the high cost of deploying scanning radars on satellites and their limited swath width means that most satellite-based precipitation observations still rely heavily on PMW data (Bauer et al., 2005; Guilloteau et al., 2018). PMW, though less precise than radars by only gathering two-dimensional (2D) radiance, provides a cost-effective and practical alternative. They offer extensive global coverage with wider swaths, making them essential for continuous monitoring of precipitation. Moreover, their multi-channel capabilities allow for an indirect inference of vertical precipitation profiles, thereby partially compensating for their inherent limitations (Guilloteau and Foufoula-Georgiou, 2020). However, accurately correlating the 2D passive data with 3D radar measurements involves complex nonlinear relationships that are challenging to decipher using traditional methods. The emergence of deep learning has introduced new possibilities for addressing these challenges (Zhu et al., 2017). By leveraging extensive observational datasets, deep learning models can learn to map the complex relationships between passive microwave measurements and active radar data, enabling the reconstruction of detailed precipitation structures.

Prior studies have applied deep learning techniques to reconstruct radar reflectivity from various satellite data sources. Many have focused on using geostationary satellite imager data to approximate ground-based radar composite reflectivity (Hilburn et al., 2021; Duan et al., 2021; Yang et al., 2023). For instance, Sun et al. (2021) employed a novel U-Net model with Fengyun-4A observations to produce radar composite reflectivity factor (RCRF) maps. Their results closely matched precipitation patterns inferred from IMERG data, achieving RMSE as low as 1.2–2.4 dBZ. In addition, several studies have used optical and infrared imagery to infer vertical cloud structures (Haynes et al., 2022; Wang et al., 2023a, b). For example, Brüning et al. (2024) combined MSG SEVIRI imagery with CloudSat 2D radar reflectivities using a Res-UNet model to generate 3D cloud structures with an RMSE of 2.99 dBZ. Although their approach captures general cloud distributions, it tends to oversimplify complex, multi-level cloud systems.

In contrast, relatively few studies have leveraged passive microwave (PMW) radiances to reconstruct radar reflectivity profiles. Yang et al. (2024) pioneered this approach by mapping relationships between GPM Microwave Imager (GMI) data and DPR reflectivities using a Hybrid Deep Neural Network (HDNN). By incorporating polarization differences and auxiliary temperature profile variables, they reduced reconstruction errors (below 4 dBZ across all altitudes) and improved accuracy near the melting layer. However, their analysis did not explicitly assess the impact of polarization differences, and their work focused exclusively on oceanic precipitation, constrained by GMI's limited channel set (Turk et al., 2018).

Building on these foundations, our study employs data from the newly launched FY-3G satellite's MWRI-RM and PMR instruments. The MWRI-RM's dual oxygen absorption channels at 50 GHz and 118 GHz provide critical temperature profile information, enabling improved representation of land-based precipitation and more accurate melting layer characterization. Moreover, incorporating Tb polarization differences may refine reflectivity reconstruction by enhancing the model's ability to discriminate between precipitation types and to capture precipitation structures in greater detail (Gong and Wu, 2017; Geer et al., 2021).

In this work, we apply deep learning techniques to reconstruct three-dimensional precipitation structures over oceans, land, and in non-precipitating conditions, fully utilizing FY-3G/MWRI-RM's extensive channel capabilities. We specifically focus on evaluating how oxygen absorption channels and polarization differences influence reconstruction performance. Our goal is to improve the accuracy and reliability of precipitation measurements derived from combined PMW and radar observations, thereby enhancing high-impact weather monitoring and forecasting.

## 2 Data and Methodology

### 2.1 Data Source and Characteristics

The primary data utilized in this study was acquired from the FY-3G precipitation satellite. Operating in a unique low-inclination, non-sun-synchronous, inclined orbit at a nominal altitude of 407 kilometers and an inclination of $50°\pm1°$, the satellite covers the global mid-to-low latitude regions (Fig. 1a). It is equipped with two important precipitation observation payloads which are the Ka/Ku Dual-frequency Precipitation Radar (PMR) and the Microwave Radiance Imager-Rainfall Measurement (MWRI-RM). These instruments are crucial for monitoring catastrophic weather systems like typhoons and heavy rainfall.

PMR: Serving the core payload of FY-3G, which provides 3D structural data on precipitation systems, the PMR is a single-polarization one-dimensional (1D) phased-array radar with a cross-track scanning mechanism, featuring a scan interval of $0.7°$ (Gu et al., 2022). It covers a ground swath width of 303 kilometers and offers a horizontal resolution of 5 kilometers at the nadir and a range resolution of 250 meters with a sampling interval of 50 meters (1b, c).

MWRI-RM: A significant payload of the FY-3G satellite, the MWRI-RM has 17 frequency points ranging from 10.65 GHz to 183 GHz, which includes nine dual-polarized channels in the 10-89 GHz spectrum and twelve oxygen absorption sounding channels around the 50 GHz and 118GHz, totaling 26 channels with spatial resolutions ranging from 5 to 25 kilometers. Detailed channel information for the MWRI RM can be found in Table 1. Employing a conical scanning regime with imaging channels and sounding channels having incidence angles of $53°\pm1°$ and $50°\pm1°$ respectively, MWRI-RM has an effective observation swath of 800 kilometers, as illustrated in Fig. 1b. This instrument captures passive microwave radiation from the Earth's surface, yielding information on precipitation, atmospheric water vapor, cloud liquid content, path-integrated liquid water thickness, melting layer height and thickness, sea surface wind speed, and more (Zhang et al., 2023a).

In this study, we also utilized the dual-polarization radar from the China Next Generation Weather Radar (CINRAD/SA) network for comparison with the land precipitation reflectivity reconstruction results. The CINRAD/SA products have a radial distance resolution of 250 m and an azimuthal resolution of 1 degree. The Volume Coverage Pattern 21 (VCP21) scan mode was selected, which sweeps 9 elevation angles of 0.5, 1.5, 2.4, 3.4, 4.3, 6.0, 9.9, 14.6, and 19.5 degrees in 6 minutes (Teng et al., 2023). Based on the interpolation method used in Xiao and Liu (2006), we transformed the radar data from spherical coordinates into a unified Cartesian coordinate system, creating a 3D grid. The radar data were then stitched and organized to generate the constant altitude plan position indicator (CAPPI) data used in this study.

The Level 1 products of PMR and MWRI-RM can be downloaded from the FENGYUN Satellite Data Center website (https://satellite.nsmc.org.cn)

**Table 1.** Detailed specifications of MWRI-RM.

| No. | Central frequency (GHz) | Polarization | IFOV (km × km) | NE$\Delta$T |
|-----|-------------------------|--------------|----------------|-------------|
| 1 | 10.65 | V, H | 21×35 | 0.5 K |
| 2 | 18.7 | V, H | 14×23 | 0.5 K |
| 3 | 23.8 | V, H | 13×21 | 0.5 K |
| 4 | 36.5 | V, H | 9×15 | 0.5 K |
| 5 | 50.30 | V, H | 7×11 | 0.5 K |
| 6 | 52.61 | V, H | 7×11 | 0.5 K |
| 7 | 53.24 | V, H | 7×11 | 0.5 K |
| 8 | 53.75 | V, H | 7×11 | 0.5 K |
| 9 | 89.0 | V, H | 5×8 | 0.5 K |
| 10 | 118.7503±3.2 | V | 4×7 | 0.8 K |
| 11 | 118.7503±2.1 | V | 4×7 | 0.8 K |
| 12 | 118.7503±1.4 | V | 4×7 | 0.8 K |
| 13 | 118.7503±1.2 | V | 4×7 | 0.8 K |
| 14 | 165.5±0.75 | V | 4×6 | 0.8 K |
| 15 | 183.31±2.0 | V | 4×7 | 0.8 K |
| 16 | 183.31±3.4 | V | 4×7 | 0.8 K |
| 17 | 183.31±7 | V | 4×7 | 0.8 K |

## 2.2 Data Preprocessing

During the data preprocessing phase, the spatial positions of MWRI-RM data were first unified. Due to the angular discrepancy in the observational geometry of the MWRI-RM's two arrays of feedhorns (S1 and S2), we applied the nearest neighbor method to adjust the spatial positioning of these two sets of data for consistency. Polarization Difference (PD) is introduced as a key analytical variable, calculated from the difference between the brightness temperatures (Tbs) of vertically and horizontally polarized channels (PD = Tb_V - Tb_H). This parameter aids in revealing the shape and size distribution of hydrometeors, providing unique perspectives on precipitation processes (Geer et al., 2021). For the MWRI-RM, polarization differences were derived from nine channel pairs across frequencies: 10.65, 18.7, 23.8, 36.5, 50.30, 52.61, 53.24, 53.75, and 89 GHz. Each frequency channel offers distinctive insights into atmospheric conditions and hydrometeor characteristics.

In processing the PMR radar reflectivity profiles, special attention was given to mitigating the influence of ground clutter, PMR scan interval of 0.7°which escalates with incidence angle. Reflectivity profiles with local zenith angles less than 2° were

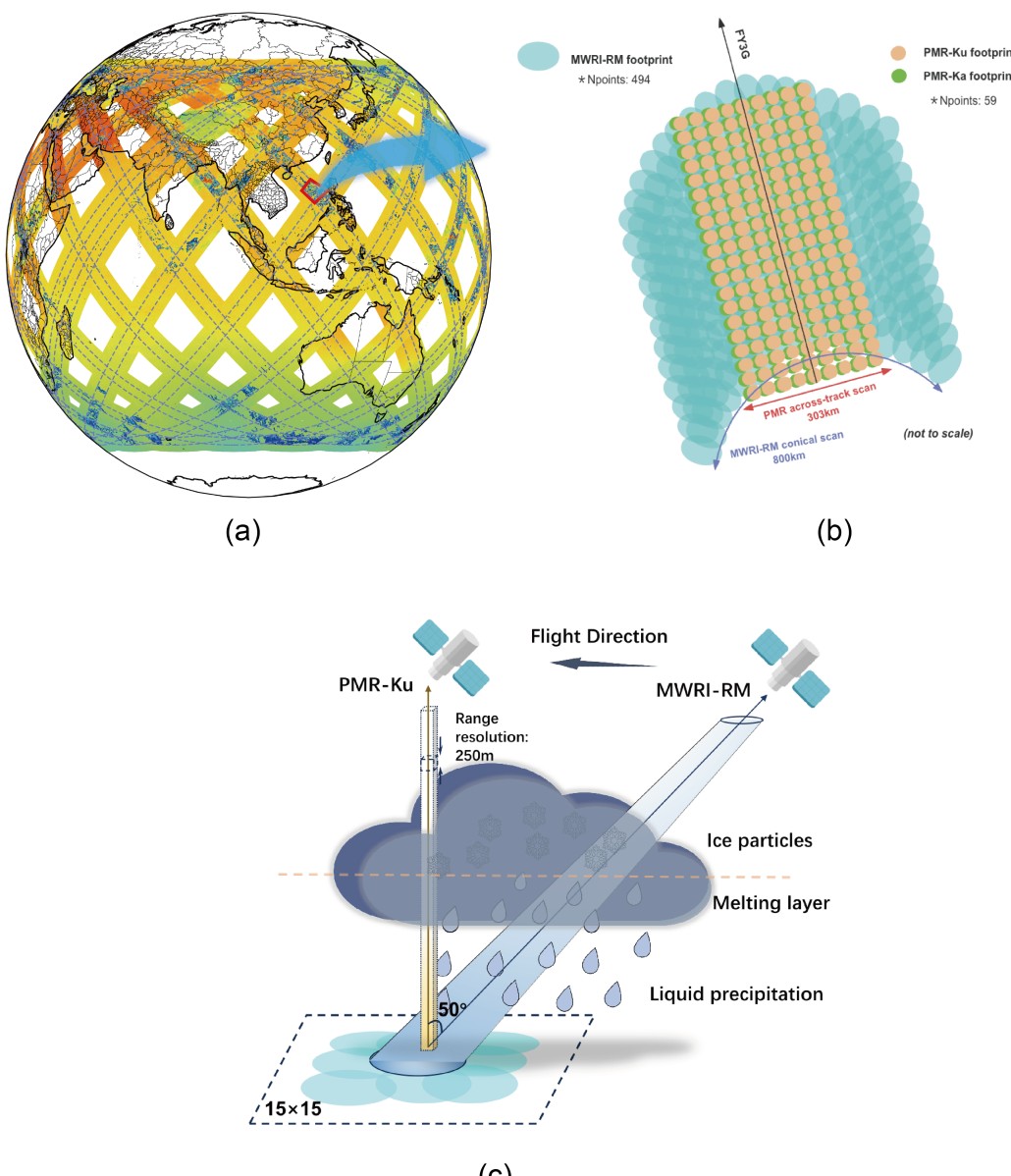

**Figure 1.** (a) The brightness temperatures from the MWRI-RM's 52.61 GHz vertical polarization channel and PMR's reflectivity at approximately 4 km above ground, recorded on July 31, 2023. The blue dashed line indicates the PMR scanning swath. (b) The viewing geometry of the MWRI-RM and PMR, with footprint sizes and numbers that do not correspond to actual observations but represent a conceptual example. 'Npoints' refers to the number of scanning points along each scan line. Adapted from Fig 1 in Turk et al. (2021). (c) Schematic representation of viewing volumes of MWRI-RM and PMR in the atmosphere, illustrating the patch extraction technique employed to the matched footprint of MWRI-RM.

selectively analyzed to minimize the interference from ground clutter, which affects altitudes up to approximately 1.1 km. Based on the 2° incidence angle criterion, five cross-track scan points per scan line meet this requirement. This approach is based on the observation that ground clutter's detrimental effects on radar measurements intensify with greater zenith angles, thus necessitating an angular threshold to uphold data quality (Kubota et al., 2016; Shimizu et al., 2023). To balance the number

of clear-sky and cloudy pixels, we only selected radar bins below 8 km, considering the cloud tops of most precipitation clouds are below this altitude (Yang et al., 2024). This helps model capture the essential data while minimizing noise from higher altitudes that might not be as relevant for precipitation analysis. The final reflectivity profiles dimension encompasses 138 bins ranging from 1.1 to 8 km. Bins impacted by sidelobe and ground clutter were marked as invalid (NaN) to prevent their potential negative influence on the reconstruction outcome. Radar bins affected by noise were assigned a baseline value of 10

dBZ, supported by statistical evidence indicating that precipitation echoes typically exceed 12 dBZ. This selected baseline value helps distinguish between non-precipitating and precipitating conditions without significantly altering the overall distribution of reflectivity values, thereby maintaining the integrity of the reconstruction process. The filtering of noise, sidelobe clutter, and ground clutter was performed using the quality flag data (flagEcho) provided in Level 1 products of PMR.

The subsequent step in data preprocessing involved aligning the MWRI-RM data with PMR observations. This alignment

was achieved by matching each MWRI-RM observation point to the nearest PMR footprint in space, ensuring a unique correspondence. Additionally, only matches with a scan time difference of less than 80 seconds were considered (Yang et al., 2024). Considering the different scanning modes of the PMR (cross-track scanning) and MWRI-RM (conical scanning), there are inherent differences in the spatial coverage of their observations (Fig. 1b). Although we performed nearest-neighbor matching for the ground footprints, the actual atmospheric volumes observed by each instrument are not identical. To best match the

atmospheric conditions observed by both instruments, a patch extraction technique was employed, centered on the matched MWRI-RM footprint (Fig. 1c). This ensures that the atmospheric column observed by the PMR is always intersected by multiple MWRI-RM scanning beams. Consequently, the MWRI-RM Tb patch encompasses the atmospheric state information of the column observed by the PMR. This approach aligns the data from the two sensors more effectively, enhancing the model's ability to accurately interpret and utilize these observations. Our preliminary studies have indicated that a patch size of 15×15

offers the most optimal reconstruction performance (Yang et al., 2024). This choice allows for the inclusion of sufficient relevant atmospheric information and ensures effective observation matching. Consequently, this study utilizes exclusively patches of this specified size for analysis.

Given the natural scarcity of precipitation events compared to non-precipitation conditions, an undersampling of non-precipitation samples was performed. This procedure aimed to balance the quantity of precipitation and non-precipitation

samples, providing a more equitable dataset for model training and evaluation. Here, the precipitation flag (flagPrecip) in the PMR data was used to distinguish between precipitation and non-precipitation samples. The operationally calibrated MWRI-RM Level 1 data and PMR Level 1 data have been available since 23 October 2023. The data from 23 October to 30 November 2023 were used for model training. After preprocessing, a total of 838,591 sample pairs were generated, including 421,090 non-precipitation samples and 417,501 precipitation samples, with 792,099 ocean samples and 46,492 land samples. The dis-

tinction between "ocean" and "land" samples was made using the "LandSurfaceType" variable provided in the Level 1 products

of PMR. These sample pairs were divided into training and validation subsets, with 80% of the samples allocated for training and the remaining 20% reserved for validation. Additionally, data from 1 December to 11 December 2023 were used for model testing, yielding a total of 288,892 samples, including 144,478 non-precipitation samples and 144,414 precipitation samples, with 274,429 ocean samples and 14,463 land samples. All inputs were standardized according to the following equation:

$$x' = \frac{x - \mu}{\sigma} \tag{1}$$

where $x$ represents the original input value, and $x'$ is the standardized value. $\mu$ is the arithmetic mean, and $\sigma$ is the standard deviation. This normalization is applied independently to each channel using the mean and standard deviation. Additionally, since the reflectivity profiles exhibited a skewed distribution, a logarithmic transformation was applied. This preprocessing step is essential for reducing the impact of outliers and ensuring that the model learns from a more uniform dataset, thereby enhancing the reliability and accuracy of the predictions. The overall preprocessing flow is summarized in Fig. 2.

## 2.3 Deep Learning Model Architecture

In our investigation, we adopt a deep learning architecture that diverges from the mixed model approach reported by Yang et al. (2024). Unlike their study, which explored the benefits of a combined CNN-MLP approach for integrating Tb patches from multiple channels and ancillary temperature profile data, our model emphasizes the potency of CNNs in discerning the vertical structure of atmospheric temperature directly from the advanced sounding channels, particularly those near the oxygen absorption bands. These sounding channels are sensitive to the temperature profile, thus potentially reducing the necessity for separate profile information processing.

To critically evaluate the influence of various channel configurations and feature inputs on the model's ability to reconstruct radar reflectivity, we orchestrated a series of controlled experiments as shown in Fig. 2. The baseline experiment (Ex14) excludes the oxygen absorption channels, using 14 input channels (out of the total 26, excluding 12 oxygen absorption channels) to assess the model's basic capability without temperature information. The full channel experiment (Ex26) incorporates all 26 input channels, including the oxygen absorption bands, providing additional temperature profiles to improve precipitation detection and the delineation of melting layers. Building on this, the polarization difference enhanced experiment (Ex35) adds Tb polarization difference data, increasing the input to 35 channels (26 channels plus 9 polarization difference sets), to enhance the model's ability to distinguish precipitation types and capture finer structural details. With a consistent architecture across experiments, this setup enables a rigorous evaluation of how extended MWRI-RM channel information influences the model's performance in precipitation reconstruction.

The Detailed specifications of the CNN model configuration are delineated in Table 2. In Table 2, $C$ denotes the number of input channels. For experiments Ex14, Ex26, and Ex35, $C$ is 14, 26, and 35, respectively. The architecture of the CNN employs a consistent structure across experiments, and the term B.R.D represents a sequence of three layers: Batch Normalization, Rectified Linear Unit (ReLU) activation layer, and Dropout. For all experiments, dropout is applied with a probability of $p = 0.2$.

**Table 2.** Architectural Details of CNN Models and Total Parameters.

| Layer Type | Parameters (Weights + Bias) | Output Shape | Total Parameters |
|---|---|---|---|
| Input | - | $C \times 15 \times 15$ | - |
| Conv (3×3) | $C \times 32 \times 9 + 32$ | $32 \times 15 \times 15$ | $288C + 32$ |
| B.R | $2 \times 32$ | $32 \times 15 \times 15$ | 64 |
| MaxPooling (2×2) | - | $32 \times 7 \times 7$ | - |
| Dropout | - | $32 \times 7 \times 7$ | - |
| Conv (3×3) | $32 \times 64 \times 9 + 64$ | $64 \times 7 \times 7$ | 18,496 |
| B.R.D | $2 \times 64$ | $64 \times 7 \times 7$ | 128 |
| Conv (3×3) | $64 \times 128 \times 9 + 128$ | $128 \times 7 \times 7$ | 73,856 |
| B.R.D | $2 \times 128$ | $128 \times 7 \times 7$ | 256 |
| GlobalAveragePooling | - | 128 | - |
| FC | $128 \times 256 + 256$ | 256 | 33,024 |
| FC | $256 \times 138 + 138$ | 138 | 35,466 |
| **Total** | - | - | **162,186** |

Model training was conducted using eight A800 GPUs, with a batch size of 512, 100 epochs, and the Adam optimizer. The learning rate was initialized at $1 \times 10^{-3}$ and decayed using the InverseTimeDecay schedule. Early stopping was employed with a patience of 20 epochs to prevent overfitting, and the best weights were restored. The loss function used was Sum of Squared Errors (SSE), defined as the sum of squared differences between true and predicted values. This choice was motivated by the specific characteristics of radar reflectivity reconstruction tasks. Reflectivity profiles often include sparse but crucial high values in precipitation regions, embedded within a majority of lower background values. SSE's quadratic nature amplifies the contribution of larger errors, effectively increasing the weight of discrepancies in high-reflectivity precipitation regions. This ensures that the model prioritizes the accurate reconstruction of physically significant regions associated with precipitation, while maintaining overall reconstruction fidelity across the entire profile. This training protocol was applied consistently across all experiments to ensure comparability and reproducibility of results. To account for the influence of random processes, three independent training runs were conducted for each model, each initialized with a different random seed. Fig. A1 illustrates the average training and validation loss curves across these runs. The results show that both training and validation losses consistently decreased at a uniform rate across all trials, with no signs of divergence or rebound. This indicates that the model avoided overfitting, demonstrating its robustness and effectiveness during training.

## 2.4 Evaluation metrics

To quantify the reconstruction accuracy of the model, we employed three statistical indices: Mean Bias Error (MBE), Standard Deviation (STD) of the predictive error, and Root Mean Square Error (RMSE). Each metric offers a distinct dimension of the model's performance.

The MBE is indicative of the average bias in the model's predictions, where a value of zero represents a perfect bias-free model. It is computed as follows:

$$\text{MBE} = \frac{1}{n} \sum_{i=1}^{n} (R_i - O_i) \tag{2}$$

Where $R_i$ is the reconstructed value, $O_i$ is the observed value, and $n$ is the number of data points. This metric reveals whether the model tends to overpredict or underpredict the observed values, providing insight into systematic deviations.

The STD of the predictive error measures the spread of these errors around the mean error, indicating the variability within the dataset:

$$\text{STD} = \sqrt{\frac{1}{n-1} \sum_{i=1}^{n} (R_i - O_i - \text{MBE})^2} \tag{3}$$

Here, the term $R_i - O_i - \text{MBE}$ represents the deviation of each individual reconstruction error from the MBE. The STD elucidates the volatility in reconstruction performance and identifies the degree to which individual reconstructions deviate from the average bias.

Finally, the RMSE assesses the magnitude of the error, penalizing larger discrepancies more severely, thus highlighting the model's precision:

$$\text{RMSE} = \sqrt{\frac{1}{n} \sum_{i=1}^{n} (R_i - O_i)^2} \tag{4}$$

The RMSE is especially useful as it relates directly to the data's scale, offering an unambiguous interpretation of the model's predictive capability.

By utilizing these indices, we can deliver a rigorous and comprehensive evaluation of our model's predictive accuracy, providing a balanced overview of both central tendency and variability in the model's performance. This multifaceted approach ensures a robust assessment, crucial for validating the model's applicability to operational forecasting scenarios.

## 3 Results

### 3.1 Reconstruction Performance and Evaluation

In this section, we conduct a comprehensive evaluation of the performance of three experiments—Ex14, Ex26, and Ex35—using 288,892 test samples. Fig. 3 illustrates the variations in MBE, RMSE, and STD across different height levels for four scenarios: oceanic precipitation, land precipitation, coastal precipitation, and non-precipitation under the conditions of the three experiments. Additionally, Table 3 provides detailed numerical values for evaluation metrics under different scenarios for each of the

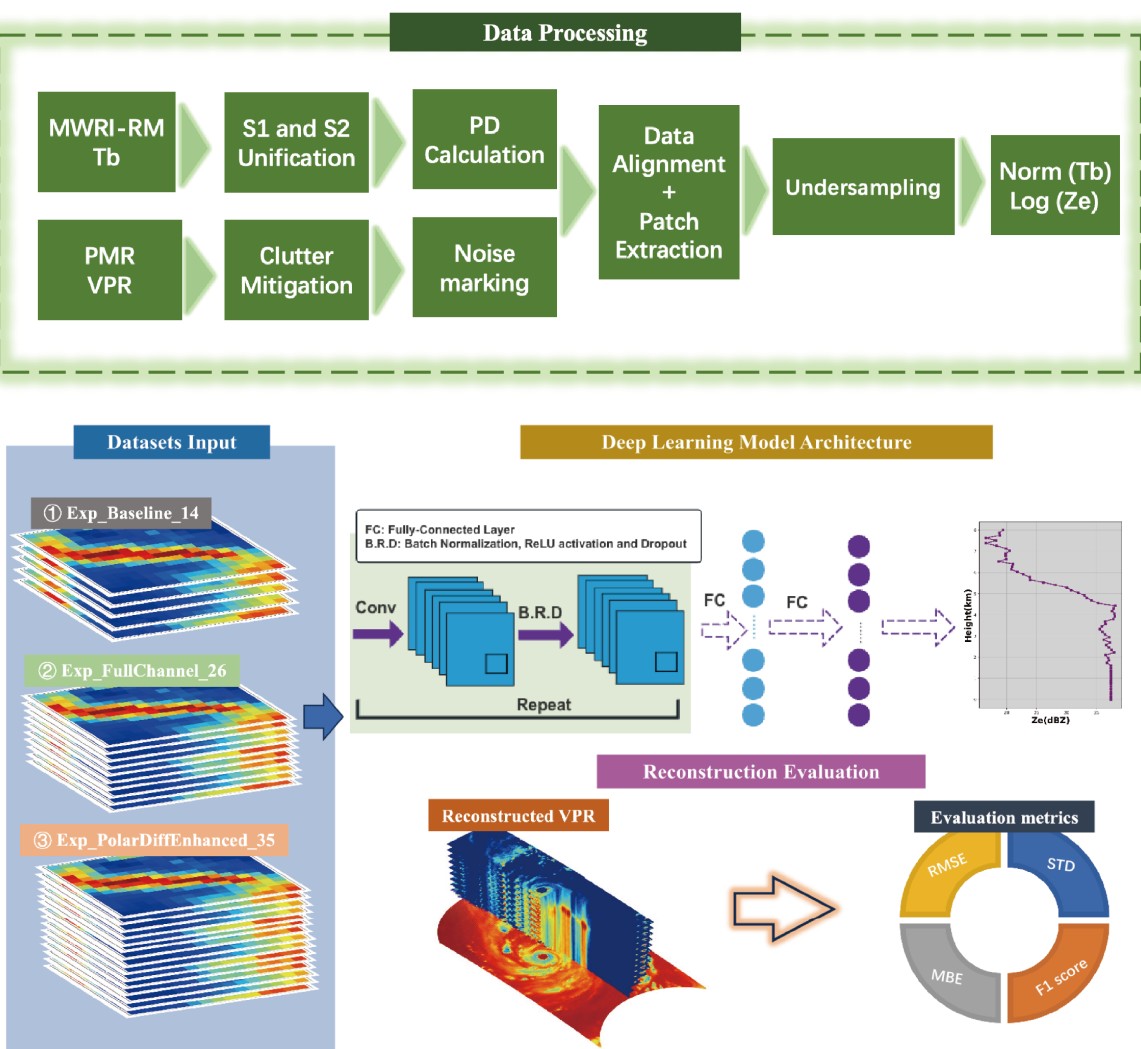

**Figure 2.** Diagram of Data Processing Workflow, Model Architecture, and Evaluation Metrics.

three experiments, offering a thorough quantitative analysis of model performance. The results presented in both the figure and table reflect the average performance across three independent training runs for each experiment, ensuring robustness against variability introduced by random initialization.

A general trend observed is the reduction of errors with increasing altitude. However, the errors at approximately 4–5 km, a range where the melting layer often occurs (Hu et al., 2024), remain significant. This highlights a crucial area for model refinement, particularly in adjusting to phase changes in precipitation. The comparison between oceanic and land precipitation scenarios reveals greater reconstruction errors over land. This discrepancy is largely attributed to the interference of land surface emissivity at low-frequency channel bands, which compromises the channels' ability to convey accurate precipitation

**Table 3.** Evaluation Metrics for Three Experiments Across Different Precipitation Scenarios. The table provides detailed numerical values for oceanic precipitation, land precipitation, coastal precipitation, and non-precipitation conditions for each experiment.

| Experiments | Scenarios | Region | Evaluation Metrics | | |
|---|---|---|---|---|---|
| | | | RMSE | MBE | STD |
| Ex14 | Precipitation | Ocean | 3.30 | 0.57 | 3.25 |
| | | Land | 4.60 | 0.90 | 4.49 |
| | | Coastal | 4.02 | 0.82 | 3.94 |
| | Non-Precipitation | Ocean | 0.41 | -0.08 | 0.40 |
| | | Land | 0.98 | -0.19 | 0.96 |
| | | Coastal | 0.58 | -0.13 | 0.56 |
| Ex26 | Precipitation | Ocean | 3.18 | 0.52 | 3.13 |
| | | Land | 4.35 | 0.62 | 4.31 |
| | | Coastal | 3.80 | 0.69 | 3.74 |
| | Non-Precipitation | Ocean | 0.37 | -0.05 | 0.36 |
| | | Land | 0.62 | -0.14 | 0.60 |
| | | Coastal | 0.52 | -0.10 | 0.51 |
| Ex35 | Precipitation | Ocean | 3.13 | 0.48 | 3.10 |
| | | Land | 4.35 | 0.58 | 4.31 |
| | | Coastal | 3.77 | 0.64 | 3.72 |
| | Non-Precipitation | Ocean | 0.36 | -0.05 | 0.35 |
| | | Land | 0.63 | -0.14 | 0.61 |
| | | Coastal | 0.50 | -0.10 | 0.49 |

information (Munchak and Skofronick-Jackson, 2013). Such limitations underscore the challenges faced by the model in land precipitation scenarios due to the reduced effectiveness of low-frequency channel observations in these environments.

The coastal precipitation scenario presents an intermediate case combining aspects of both land and ocean surfaces. In practice, this scenario emerges because the passive microwave brightness temperatures used for the reconstruction cover a 15×15 field of view (FOV) across multiple channels, which may encompass both oceanic and terrestrial footprints within the same retrieval domain. The coastal scenario refers to all observations where the passive microwave FOV includes both land and ocean components, regardless of whether the corresponding radar observations are positioned over land or ocean. As a result, certain pixels or sub-regions within this FOV inherit mixed-surface characteristics, leading to "coastal" conditions where parts of the passive channels are affected by land. This heterogeneous surface environment introduces an intermediate complexity

level. Errors in coastal precipitation scenarios are somewhat higher than those seen over pure ocean but noticeably lower than those over land, reflecting a gradual degradation in retrieval accuracy as land influence grows within the 15×15 channel domain.

With the integration of dual oxygen absorption sounding channels, there is a notable improvement in the accuracy of precipitation reflectivity reconstruction, especially under land and coastal precipitation scenarios, with RMSEs reduced by 5.43% and 5.47%. This enhancement is also evident near the melting layer, underscoring the sounding channels' sensitivity to atmospheric temperature structures. This feature is instrumental in refining the model's ability to accurately map precipitation structures by leveraging the temperature profiles provided by the oxygen absorption sounding channels. The incorporation of Tb polarization difference further enhances the model's performance. However, this improvement is relatively limited, particularly in precipitation scenarios over land, as low-frequency PD signals are significantly influenced by surface properties such as soil moisture and vegetation, limiting their utility in refining precipitation signals (Paloscia and Pampaloni, 1988; Liu et al., 2014; Wang and Wang, 2022). In non-precipitation conditions, the model shows minimal error, with RMSE values generally less than 1 dBZ. The slight difference between the background value and the minimal precipitation threshold (2 dBZ) reinforces the model's capability to accurately detect the absence of precipitation.

Despite these advancements, a persistent challenge persists: all models tend to underestimate precipitation reflectivity across scenarios. This is evidenced by a generally negative mean bias error (MBE), indicating that reconstructed reflectivity values are systematically lower than observed values. Among the configurations, EX35 demonstrates the smallest underestimation, underscoring its relative advantage in mitigating this bias. However, this systematic low bias suggests potential limitations in capturing the full intensity of precipitation events, meriting further investigation.

### 3.2 Model Performance Evaluation During Extreme Precipitation Events

In this part of the paper, we focus on evaluating the model's reconstruction capabilities during two extreme precipitation events: Typhoon Khanun in 2023 and the extreme rainfall in the Beijing–Tianjin–Hebei (BTH) region in North China from July 29 to July 31, 2023 (Fowler et al., 2024). These events caused significant impacts on human life and activities, providing challenging yet informative scenarios for testing the model's precision and robustness. The challenges arise from the extreme nature of these events, which involve complex and dynamic atmospheric conditions that introduce significant variability and non-linear interactions. These factors make it difficult for the model to generalize, especially when such events exceed the range of conditions encountered during training. Although the data used for these case studies was obtained before the official release in October 2023, we were granted access to PMR observations from July 2023, including data from these two extreme events, which are valuable for evaluating the model's generalization ability in extreme conditions.

Fig. 4 and 6 provide a comparative analysis of the three-dimensional reflectivity structures of the two extreme precipitation events, across three different experimental models. The figure contrasts the observed reflectivity values (a-c) with the reconstructed data obtained from three experiments: Ex14 (d-f), Ex26 (g-i), and Ex35 (j-l). Each set of panels is organized to include three key perspectives: the 3D reflectivity structure (left column), the horizontal distribution at 5 km altitude (middle column), and the vertical cross-section along points A and B (right column). The reconstructed results presented here represent the best-

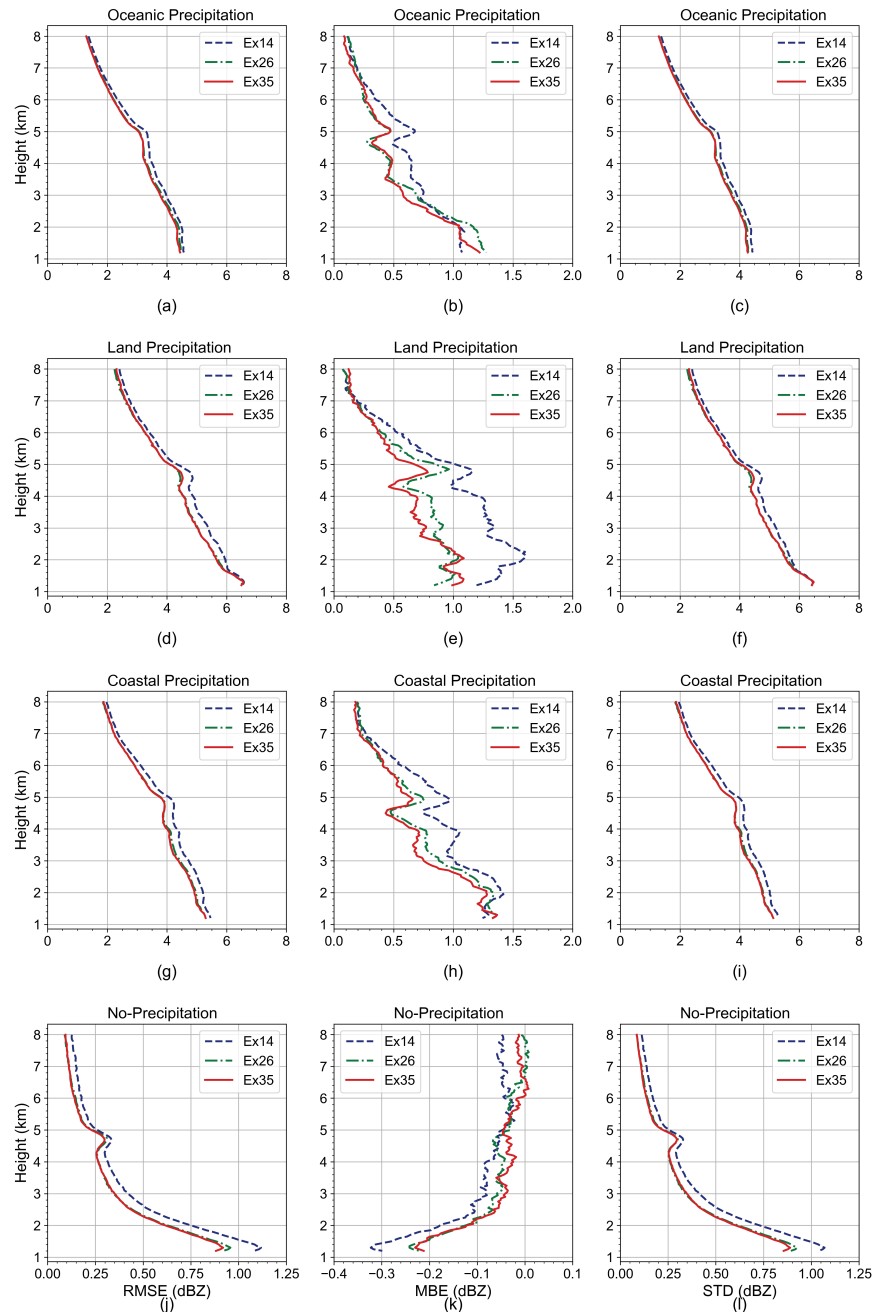

**Figure 3.** Comparison of RMSE (left), MBE (middle), and STD (right) Error Profiles for Reflectivity Reconstruction Across Different Precipitation Scenarios Using Three Experiments: Ex14 (purple line), Ex26 (green line), and Ex35 (red line). Highlighted shading indicates the position of the melting layer in each scenario.

performing outcomes from each experiment to highlight the models' optimal capabilities in capturing extreme precipitation features.

As shown in Fig. 4, the 3D reflectivity profile from PMR data revealed decreasing reflectivity values with altitude, typical of frozen precipitation forms such as snow, graupel, and frozen drops commonly found in the cores of active thunderstorms. Ground clutter limited the observation range at the lowest altitudes, but the model, trained on reflectivity profiles with incidence angles less than 2°, reconstructed a more complete reflectivity profile, minimizing ground clutter effects. This demonstrates the model's capability to reconstruct reflectivity impacted by clutter, an essential feature for accurate storm characterization. For a detailed horizontal analysis of the precipitation system, we chose a reflectivity cross-section at 5 km above ground, where the influence of ground clutter was minimal, revealing a more comprehensive view of the precipitation system. PMR data showed the highest reflectivity values located mainly to the south and east of Khanun's center, with moderate to heavy rainfall surrounding the eye and a prominent rainband to the northwest. Additionally, a vertical cross-section along a line from point A to point B (both directly beneath the satellite track) was analyzed to assess the vertical structure of the precipitation system. In this radar cross-section, a prominent bright band (BB) was observed, characterized by a peak in reflectivity. This feature typically signifies the presence of a melting layer, as it occurs where falling snow begins to transition into rain, thereby enhancing the radar return signal (Xie et al., 2024).

Analysis of the reconstruction results from different experiments showed that while all models accurately reconstructed the overall structure of the precipitation system, there were noticeable differences in detail. The Ex14 exhibited an overall weaker reflectivity reconstruction (Fig. 4d, e). The reconstructed melting layer structure was overly smooth (between 4 and 5 km) indicating a lack of detailed temperature information, which is crucial for accurately capturing this feature. For the precipitation reconstruction in this case, Ex14 showed an overall MBE of 0.81 and an RMSE of 4.91. The Ex26 provided a clearer understanding of the melting layer structure as depicted in Fig. 4i. However, it generally underestimated reflectivity values in the northwestern rainbands (26°N, 130°E) while overestimating those east of Khanun's center (23°N, 133°E) (Fig. 4d, e). Specifically, Ex26 showed an overall mean bias error (MBE) of 0.61 and a root mean square error (RMSE) of 4.87. The Ex35 results were closer to observations in detail, yet still slightly overestimated reflectivity values east of Khanun's center (23°N, 132°E) (Fig. 4j, k). Its representation of the melting layer was more accurate, closely matching actual observations (Fig. 4l). In terms of quantitative evaluation metrics, Ex35 showed an MBE of 0.58 and an RMSE of 4.86. However, as with the Typhoon Khanun event, all experiments showed some discrepancies in the position of the melting layer, typically positioned lower than observed. This is likely due to the inherent ambiguity of the radiative signals, which affects the precision and vertical detail of the reconstructions.

To provide a more quantitative and objective assessment, Fig. 5 presents scatter density plots that compare reconstructed reflectivity against the observed values for Typhoon Khanun. These plots visualize the joint distributions, with the diagonal line indicating perfect agreement. All experiments show robustness in moderate precipitation but struggles with extremes, while Ex14c exhibits the largest uncertainties.

An extreme rainfall event occurred in the Beijing–Tianjin–Hebei (BTH) region in North China from 29 July to 1 August 2023, commonly referred to as the "23·7" BTH extreme rainfall event. According to observations, the 72-hour accumulated

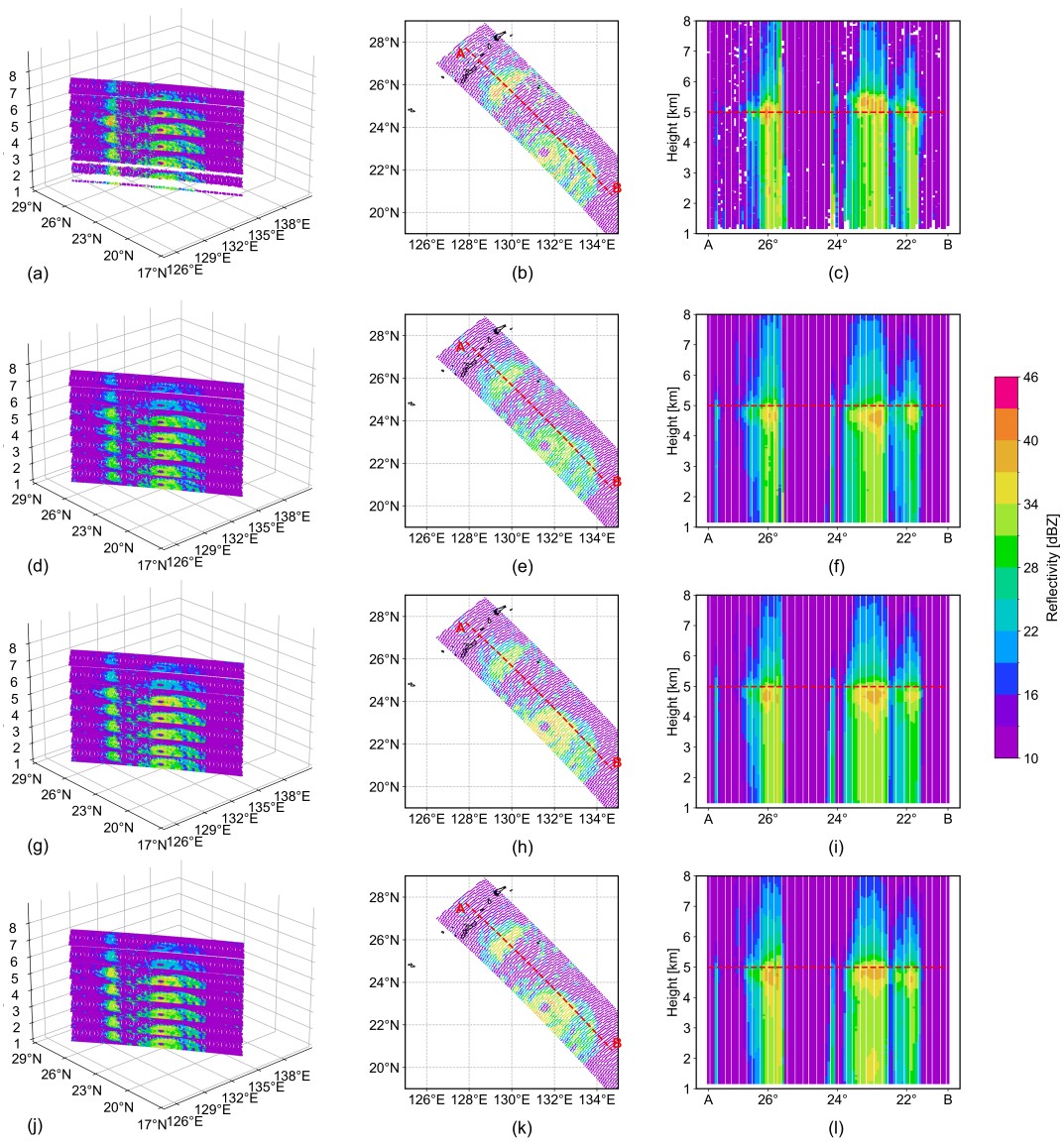

**Figure 4.** Comparative Analysis of Three-Dimensional Reflectivity Structures of Typhoon Khanun (05:30 UTC July 31, 2023) Across Three Experiments. The panels compare observed values (a-c) with reconstructed data from Ex14 (e-f), Ex26 (g-i), and Ex35 (j-l) models. Each set includes 3D reflectivity structure (left), horizontal distribution at 5km (middle), and vertical cross-section along points A and B (right).

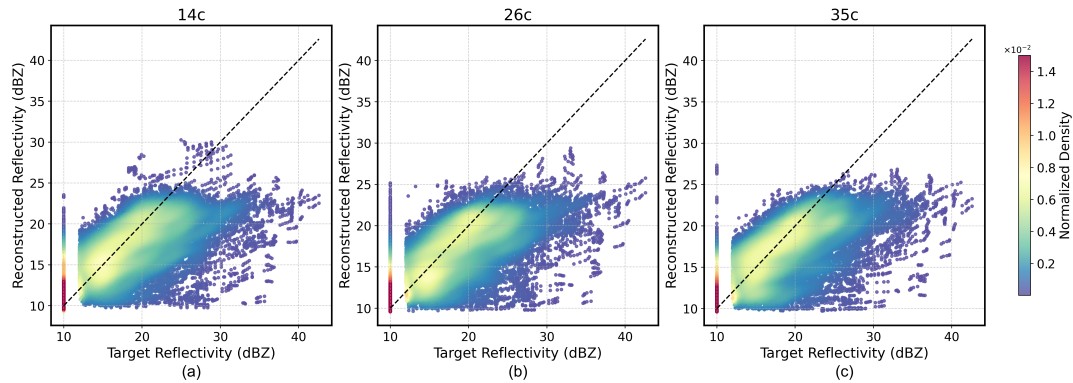

**Figure 5.** Scatter Density Comparison of Observed and Model-Reconstructed Reflectivity for Typhoon Khanun (05:30 UTC July 31, 2023) Across Three Experiments. Data point density is represented by a color scale, with warmer colors (reds, yellows) indicating higher densities and cooler colors (blues, purples) indicating lower densities. The dashed diagonal line denotes perfect agreement (1:1 line) between observations and model estimates.

precipitation from 0000 UTC on 29 July to 0000 UTC on 1 August 2023 ranged from 350 to 550 mm across southwestern Beijing, central Hebei, and Tianjin, with some localities receiving over 750 mm (Zhao et al., 2024). The FY3G satellite flew over the BTH region during this extreme precipitation event, as illustrated in Fig. 6. At 5 km altitude, PMR observations showed high reflectivity predominantly in the southern parts of Beijing (BJ, 39.5°N, 115.8°E) and central Hebei(HB, 39.1°N, 116.2°E), although the edge of the satellite track was affected by terrain, leading to missing reflectivity data in these mountainous regions. In the vertical structure analysis along a cross-section from point A to point B in southern Beijing, the bright band (BB) in this melting layer occurred around 5 km. Similar to the findings from the Khanun typhoon, all experiments reconstructed the overall precipitation system structure accurately but differed in detail. Ex14 exhibited systematic underestimation of reflectivity values, particularly failing to capture high-reflectivity regions in southern Beijing and central Hebei. This experiment also showed notable deficiencies in reconstructing the melting layer signature, suggesting that the absence of temperature information from oxygen absorption channels critically impairs vertical precipitation characterization over land. This observation aligns with previous model evaluation findings. Quantitative analysis showed Ex14 had the largest biases, with a Mean Bias Error (MBE) of 2.65 and Root Mean Square Error (RMSE) of 5.81. While Ex26 still generally underestimated reflectivity values, it demonstrated marked improvement in capturing key high-reflectivity features and produced a more realistic representation of melting layer structure compared to observations. This enhanced performance is reflected in its reduced errors (MBE=1.72, RMSE=5.23). The Ex35 configuration achieved the most accurate reconstruction, successfully retrieving high-reflectivity features in southern Beijing and producing the faithful vertical depiction of melting layer characteristics. This superior performance is quantified by the smallest errors among the three experiments (MBE=1.36, RMSE=5.14).

To further quantify model performance, we employed scatter density plots (Fig. 7). In the BTH case, Ex14 exhibited the poorest agreement with observations, characterized by a wide scatter dispersion and a pronounced systematic offset below the diagonal. Ex26 showed marked improvement, with a more symmetric density distribution concentrated closer to the 1:1

line compared to Ex14. However, it retained a tendency to underestimate reflectivity values above 25 dBZ, particularly in high-intensity precipitation regimes. Ex35's scatter density distribution is more concentrated around the diagonal, with better alignment at higher reflectivity, indicating better overall correspondence between observed and reconstructed reflectivity.

However, a notable issue in the reconstruction is the occurrence of reflectivity values below the surface in high-altitude regions, particularly areas with elevations above 1.1 km. For instance, in Fig. 7(c), which represents the observations, there are invalid values near points A and B due to the influence of mountainous terrain (elevations exceeding 1.1 km). However, in the reconstruction results, reflectivity values are still present below 1.1 km ( Fig. 6f, i, l), highlighting this notable inconsistency. This issue highlights a limitation in the model's applicability to complex terrains and mountainous regions. The potential

solutions for this issue, such as designing the loss function to account for surface constraints, incorporating terrain height as an input feature, and applying post-processing techniques, will be addressed in future work.

Based on the above evaluation, the Ex35 model demonstrates superior performance in reconstructing the horizontal distribution, vertical structure, and intensity of reflectivity for both oceanic and terrestrial precipitation. Here, we utilize the Ex35 model to reconstruct the reflectivity across the entire MWRI-RM swath to gain a more comprehensive view of precipitation

systems. The spatial distributions of brightness temperatures of the oxygen absorption band channels for two cases are shown in fig. B1 and B2, where we can clearly observe the impact of precipitation on the channel brightness temperature, with lower brightness temperature, especially in the 118 GHz channel, due to the strong effect of ice particle scattering in convective areas. Fig. 8a and c show the reconstructed reflectivity of Typhoon Khanun across the entire MWRI-RM swath. Compared to the actual observations from PMR-Ku, we can observe a very complete three-dimensional structure of the typhoon. For instance,

the well-formed spiral rainbands accompanied by intense reflectivity in the southwest of Typhoon Khanun's eyewall have been well reconstructed, a feature not evident in the PMR-Ku observations. Similarly, Fig. 8b and d present a more complete reflectivity distribution of a terrestrial precipitation system. In addition to the precipitation echoes observed by PMR-Ku over southern Beijing, central Hebei, and Tianjin (TJ, 39.2°N, 117.0°E), the MWRI-RM swath reveals precipitation echoes over northern Hebei, northern Shanxi, and parts of Inner Mongolia adjacent to Shanxi.

To validate these reconstructed reflectivities, we compare them against a composite of ground-based radar observations. Fig. 8e shows a CAPPI product compiled from 99 CINRAD/SA radars in and around the study area. Within the regions covered by ground-based radars, the spatial patterns of reflectivity and the locations of major reflectivity maxima in the reconstructed field correspond closely to the observations of ground-based radars. For instance, the model accurately reproduces the strong echoes and their approximate positions in southern Beijing and central Hebei, reflecting areas of intense precipitation similar to

those captured by the radars. This spatial alignment demonstrates that the reconstructed precipitation reflectivity distributions are consistent with ground-based radar measurements in terms of key precipitation features (e.g., high-reflectivity cores).

It is worth noting that ground-based radar coverage is limited in remote regions, such as northern Shanxi (SX, 39.8°N, 113.0°E) and Inner Mongolia (IM, 40.0°N, 112.0°E), where the radars are sparse or absent. In these areas, the reconstructed field extends beyond the radar coverage, suggesting precipitation where radars cannot confirm its presence. While this part

of the reconstruction cannot be directly validated against ground-based observations, the demonstrated consistency within the radar-covered domain increases confidence that the satellite-based retrieval is capturing realistic precipitation distributions

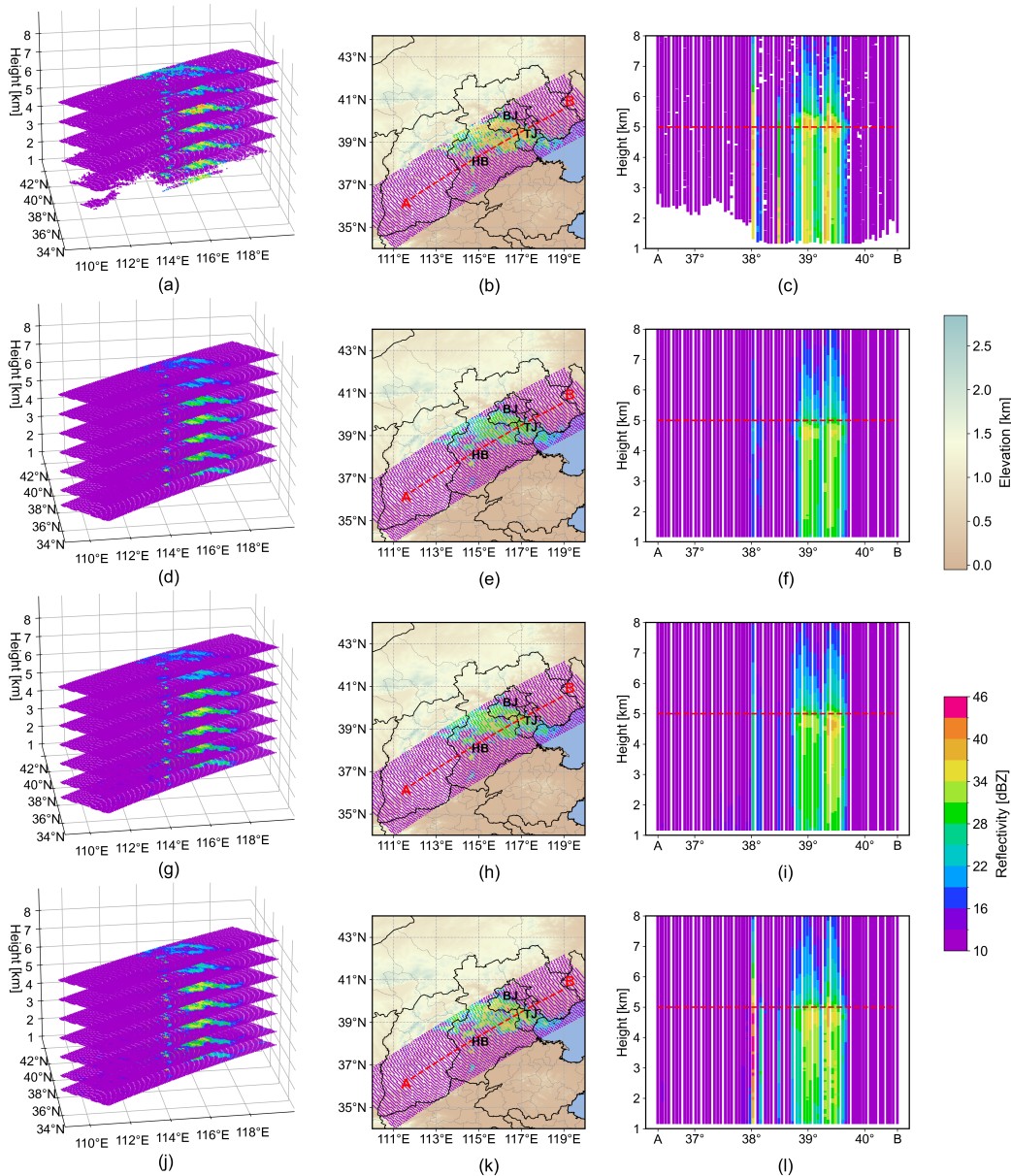

**Figure 6.** Similar to Fig. 6, but for the "23·7" BTH extreme rainfall event (23:18 UTC, July 30, 2023). Panel (b) highlights the locations of Beijing (BJ), Tianjin (TJ), and Hebei (HB) provinces, labeled in black font.

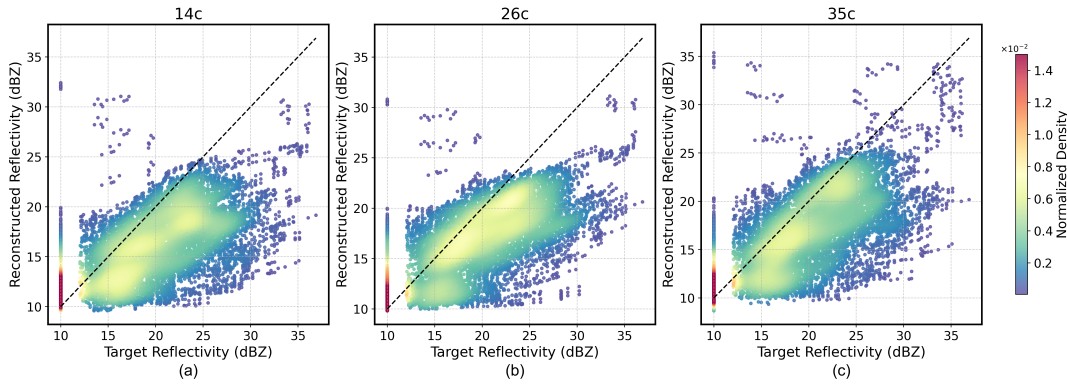

**Figure 7.** Similar to Fig. 6, but for the "23·7" BTH extreme rainfall event (23:18 UTC, July 30, 2023).

elsewhere. Hence, the Ex35 model's ability to align with radar-derived precipitation structures in well-observed areas, coupled with its broader swath coverage, indicates its potential to fill observational gaps and offer a more complete depiction of precipitation systems than ground-based radars alone.

## 4    Conclusions and discussions

The study demonstrates the significant impact of incorporating additional sounding channels and polarization differences in enhancing precipitation reconstruction models. It also introduces innovative use of analytical evaluation methods, which is critical in assessing the efficacy of new channels dedicated to precipitation detection and contributes to the ongoing advancement of remote sensing technology.

Three experiments: Ex14, Ex26, and Ex35, were evaluated, revealing the effects of dual oxygen absorption sounding channels and polarization differences on reconstruction outcomes. The findings indicate a general trend of error reduction with increasing altitude, though significant errors persist at the melting layer, highlighting a need for model refinement in adjusting to phase changes in precipitation. Reconstruction errors are greater over land compared to oceanic scenarios due to land surface emissivity interference with low-frequency channels, emphasizing the challenges in land precipitation scenarios. The coastal precipitation scenario presents an intermediate case combining aspects of both land and ocean surfaces.

The integration of dual oxygen absorption sounding channels significantly improved the accuracy of precipitation reflectivity reconstruction, especially under land and coastal precipitation scenarios, with RMSEs reduced by 5.43% and 5.47%. This improvement underscores the channels' sensitivity to atmospheric temperature structures, enhancing the model's ability to map precipitation structures accurately. Further refinement was achieved by incorporating brightness temperature polarization differences, although this enhancement had a relatively modest impact on performance across the test samples. Despite these enhancements, the model exhibited a systematic low bias in reflectivity estimation, as indicated by a generally positive MBE. In non-precipitation conditions, the model showed minimal error, with RMSE values less than 1 dBZ, reinforcing its capability to detect the absence of precipitation accurately.

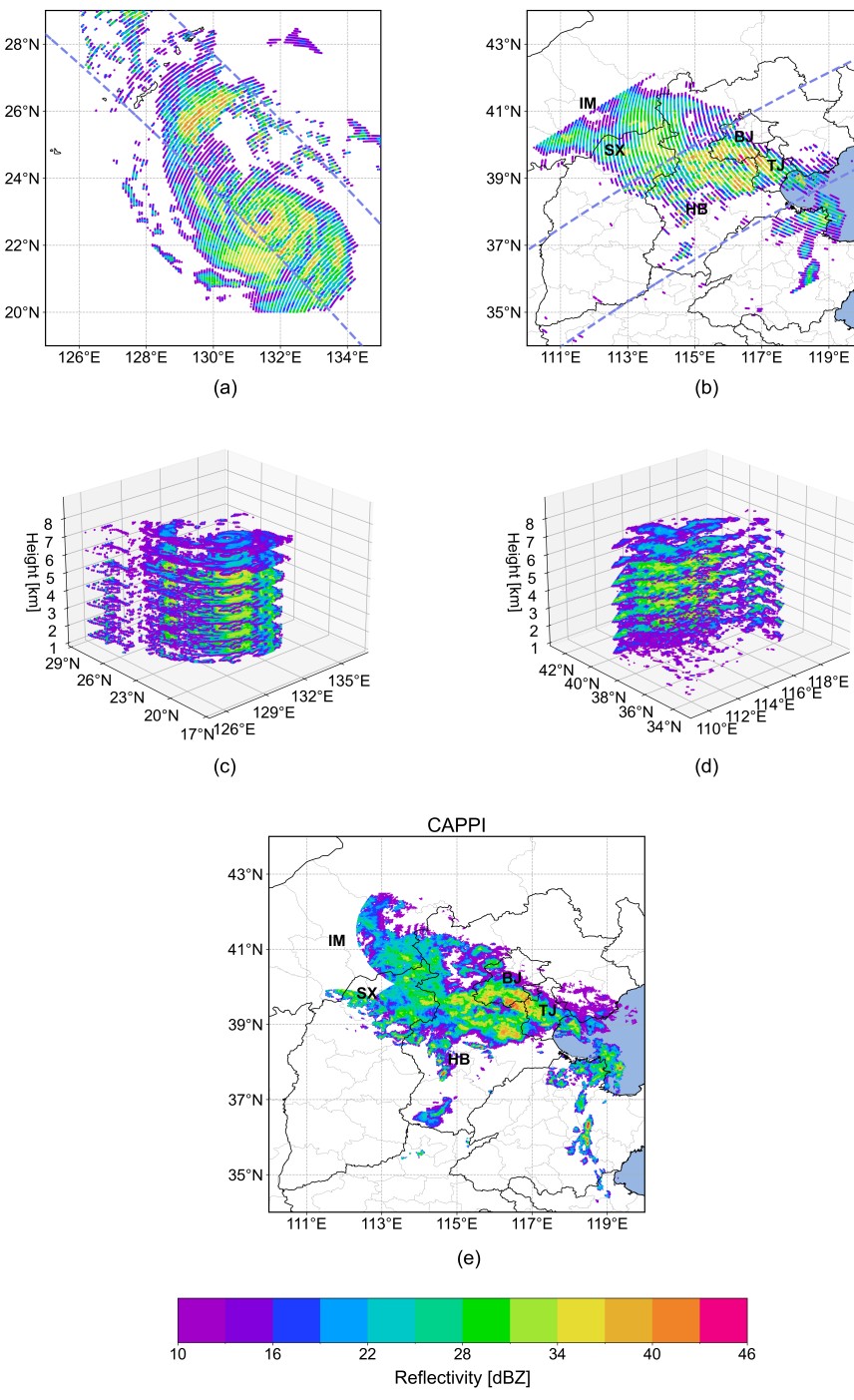

**Figure 8.** Ex35 model reconstructions of reflectivity from Typhoon Khanun (05:30 UTC July 31, 2023; left colunm) and the "23·7" BTH extreme rainfall event (23:18 UTC July 30, 2023; right colunm) within the MWRI-RM swath. (a), (b) and (e) depict the horizontal distribution at a 5 km altitude

Evaluating the model during extreme precipitation events, such as Typhoon Khanun and the extreme precipitation in Beijing,
further demonstrated its precision and robustness. The model effectively reconstructed the overall structure of precipitation systems, with Ex35 providing the most accurate detail and closely matching actual observations. This model demonstrated superior performance in reconstructing horizontal distribution, vertical structure, and intensity of reflectivity for both oceanic and terrestrial precipitation. Future work will focus on improving the model by employing a richer and more diverse training dataset, conducting more extensive validation, and demonstrating derived 3D radar reflectivity in near real-time (NRT).

However, despite these advancements, our study reveals that the reconstructed reflectivity profiles from deep learning models, while having the same vertical discretization as radar observations, do not fully achieve the level of detail and accuracy observed in direct radar measurements. This discrepancy highlights the inherent challenges in bridging the gap between passive microwave data and active radar observations, largely due to the fundamental differences in how these instruments collect data. From a theoretical standpoint, the vertical resolution of passive microwave retrievals is constrained by the limited number of
independent pieces of vertical information that can be extracted from the available channels (Turk et al., 2018). While radar instruments achieve high vertical resolution by directly measuring the backscattered signal at fine vertical intervals (Gao et al., 2017), radiometers rely on passive measurements integrated over broad weighting functions. Each frequency channel of a passive radiometer provides a certain "weighted average" of the atmospheric column, and the number of channels sensitive to distinct atmospheric layers effectively sets an upper bound on the number of independent vertical layers that can be retrieved.
For example, dual oxygen absorption channels around 50 GHz and 118 GHz can yield only a few degrees of vertical freedom for liquid water content (Han et al., 2015), far fewer than the >100 discrete vertical levels that radar measurements directly resolve. To overcome these limitations, our future work could explore integrating passive microwave observations with additional datasets, such as information from the FY-3G's onboard sensors like the Medium Resolution Spectral Imager-Rainfall Measurement (MERSI-RM), which is equipped with eight spectral channels ranging from visible to infrared wavelengths (centered
from 0.65 to 12 $\mu m$) and has a resolution of 500 meters. These independent ancillary data would enhance the spatial and physical detail available for model training, potentially improving the accuracy and detail of reconstructed precipitation structures. Furthermore, employing advanced generative models such as Generative Adversarial Networks (GANs) or diffusion models could provide innovative ways to refine the precision of precipitation reconstructions, leveraging their capacity to generate high-fidelity outputs from relatively coarse inputs (Zhang et al., 2023b). For example, Leinonen et al. (2019) demonstrated
the potential of GANs by applying a Conditional GAN (CGAN) to reconstruct two-dimensional cloud vertical structures observed by the CloudSat radar using MODIS-derived inputs. Their study highlighted the ability of CGANs to infer multilayer cloud structures and handle incomplete data, providing outputs that closely resembled radar observations. While deep learning models offer a robust framework for linking passive and active observational datasets, the quest to fully replicate the detailed observational capabilities of radar using data from passive sensors continues to pose significant scientific and technical chal-
lenges. Multi-data fusion techniques and advanced modeling approaches may hold the key to resolving these issues, offering promising avenues for future research in precipitation measurement and modeling.

# Appendix A: Training and validation loss curves

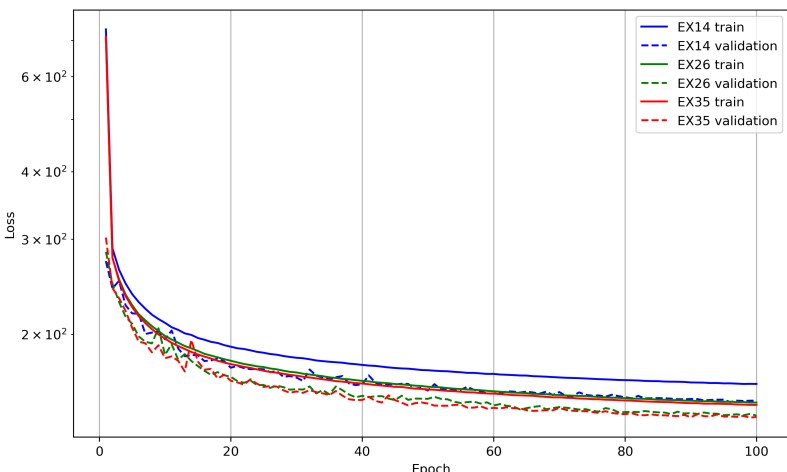

**Figure A1.** Training and validation loss curves for different input channel configurations.

# Appendix B: Spatial distributions of brightness temperatures

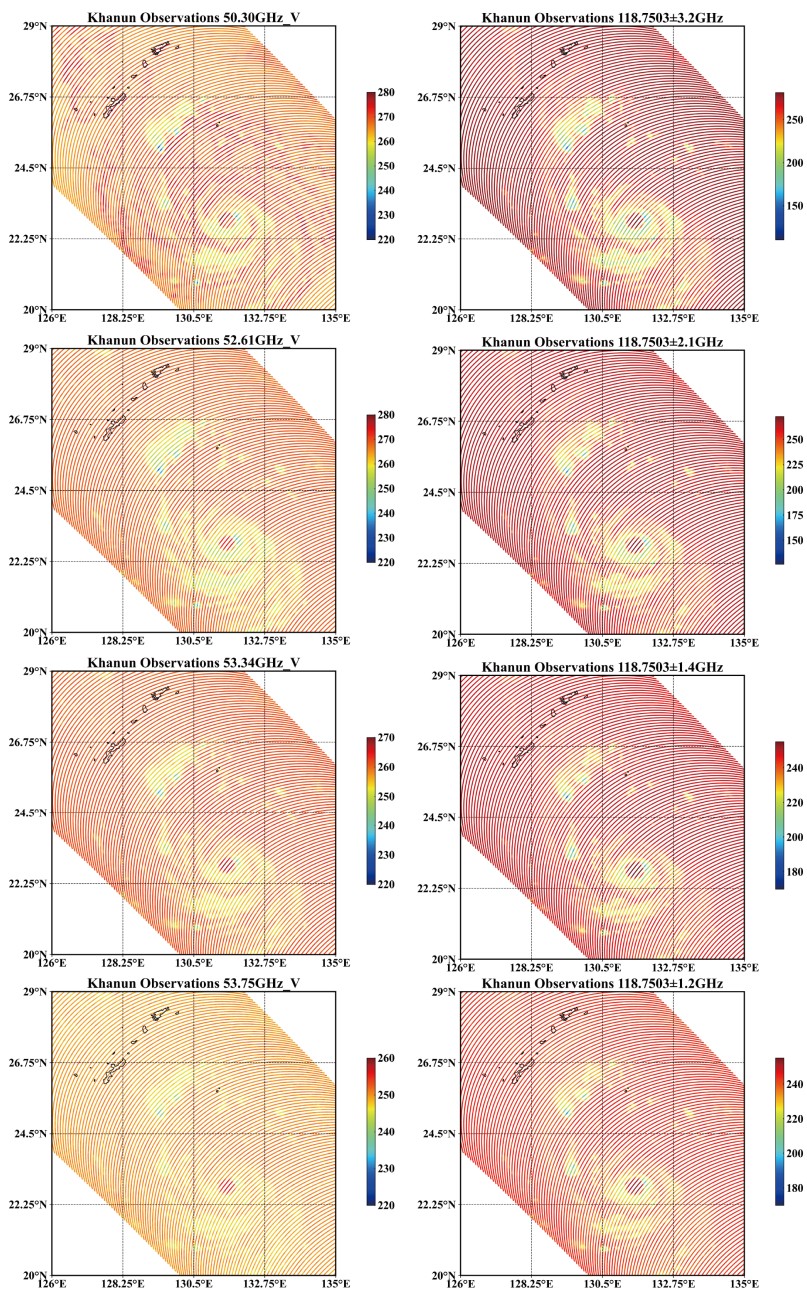

**Figure B1.** Spatial distributions of brightness temperatures observed by MWRI-RM for the oxygen absorption band channels during Typhoon Khanun (05:30 UTC, July 31, 2023).

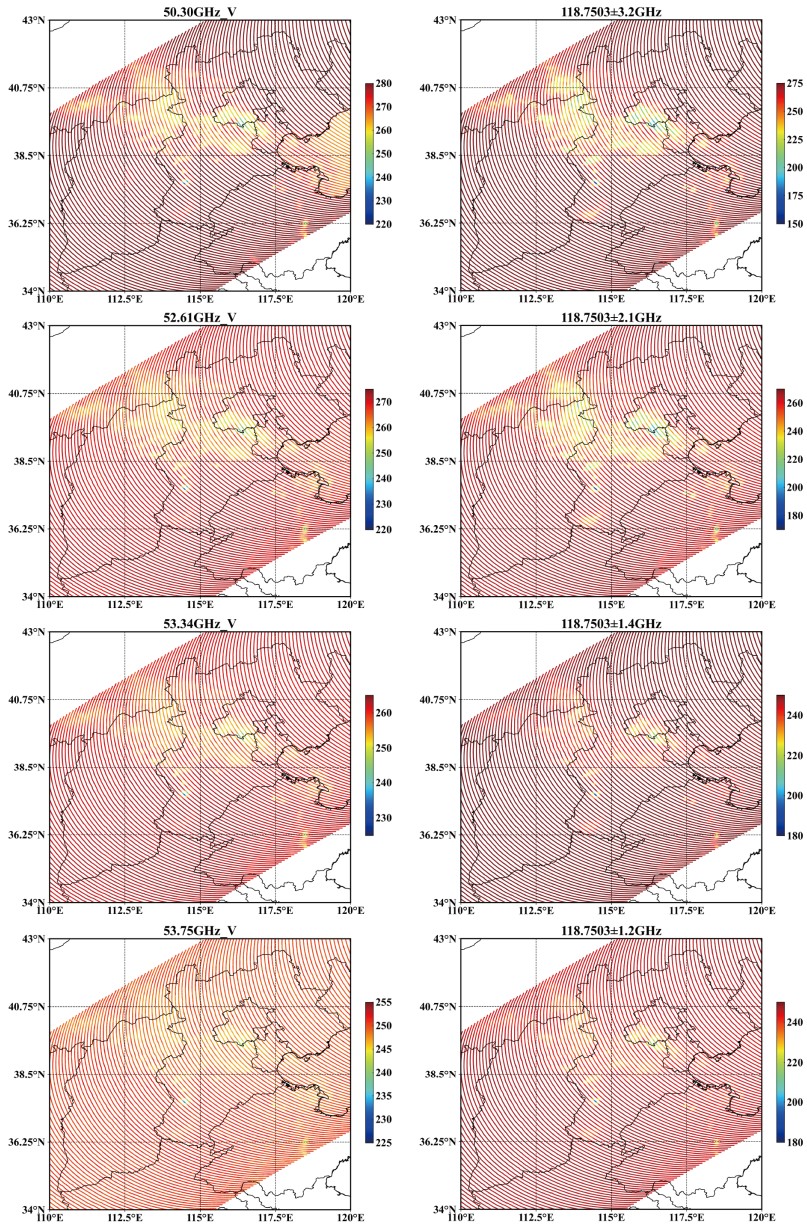

**Figure B2.** Similar to Fig. A2, but for the "23·7" BTH extreme rainfall event (23:18 UTC, July 30, 2023).

*Code and data availability.* The code used to data preprocessing and training the deep learning models are available at (https://doi.org/10.
5281/zenodo.13903700). The training datasets used for this study can be requested from the authors.

*Author contributions.* WH, YY conceived the study. YY wrote the first draft. YY and HS contributed to data processing and model training. JY processed and mapped the ground-based radar data. WH, YY, JL and ZG reviewed and revised the manuscript. WH contributed to resources. All of the authors read and approved the final manuscript.

*Competing interests.* The contact author has declared that none of the authors has any competing interests.

*Acknowledgements.* This study was jointly supported by National Key R&D Program of China(2022YFC3004004) and National Natural Science Foundation of China (42075155).

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
