# Peer review of "Reconstruction of 3D precipitation measurements from FY-3G MWRI-RM imaging and sounding channels"

_Atmospheric Measurement Techniques, 2024_

## Referee Comment (RC3)

Review Paper amt-2024-175

**Overview**

This paper investigates the possibility of using the TB observations from the Microwave Radiance Imager-Rainfall Measurement (MWRI-RM) onboard the Fenyung-3G (FY-3G) satellite to reconstruct the 3D reflectivity profiles. The algorithm is based on a deep learning approach and uses the Ku profiles observed by the Precipitation Measurement Radar (PMR) onboard FY-3G as reference truth. Three deep learning models are trained using different sets of predictors, and two case studies are analyzed.

**General Comment**

The results presented in the paper are really interesting and pave the way for new research activities. However, some aspects related to the training phase of the algorithm, in particular the selection of the training and test datasets, are not clearly explained in the text. This step plays a very important role in the development of a deep learning based algorithm. Therefore, I think that a careful revision of this part of the paper is necessary before publication.

**Specific Comment**

1) Section 2.1: I would like to suggest that this section be split into two subsections, one (L 99-103) relating to PMR characteristics, the other to MWRI-RM (L 104-114).

2) L.107: Here the authors correctly cite the paper where a table of MWRI-RM characteristics is reported. However, I would suggest that this table be added to the paper, as MWRI-RM is the instrument on which the paper is based.

3) Section 2.2: I think a summary table of the main characteristics of the dataset used in this paper (time period, number of samples, number of precipitating and non-precipitating samples, number of samples over sea and over land) would make the paper more readable.

4) L.152: It is not clear to me what the oversampling consists of. Are the same data duplicated in the dataset? I think the authors need to specify this and discuss the consequences of this choice. In particular, how can this operation affect the dataset, which is split into two sub-datasets for the training and test phases? Also, I suspect that the precipitating samples were duplicated, not the non-precipitating pixels.

5) L.155-156: I think the authors need to provide more information on how the "training" and "test" samples were selected. Is this separation based on a random process? Or by selecting observations from a defined time period? This selection plays a very important role in the development of machine learning approaches. If the two sub-datasets are created by randomly selecting the pair samples, the two sub-datasets may be highly correlated, so the statistical scores calculated over the test dataset may be not representative. In this case, the authors must recalculate the statistical indices on a test dataset independent of the training dataset - using time periods or orbits not used in training.

6) Section 2.3: Again, I would suggest that the authors add a table summarising the input datasets of the three experiments reported.

7) L.214-218: It is not clear to me how precipitating/non precipitating pixels are defined. Is it related to a reflectivity threshold? Is it the 12 dBZ threshold cited at L. 135?

8) Table 2 and Figure 5: I guess that the statistical scores and the mean profiles are calculated over the test dataset; however, this information is not clearly reported in the text. I would like to suggest that the authors specify which dataset the scores are calculated on. Again, the RMSE, STD, and MBE have been calculated only for the true positive (for precipitating samples) and for the true negative (for non-precipitating samples)? This information must be added to the text.

9) L.234-235 and Figure 5: how the melting layer height is determined?

10) L. 258-260: The reported period is outside the period of the development dataset. This makes the results more valid but must be stated in the text.

11) Figure 8: A general weakness of the paper is the lack of references to the brightness temperatures observed by the MWRI-RM, which are the input data to the deep learning algorithm. I understand that reporting TB maps can be difficult given the number of channels, but perhaps a TB map could be added to Figure 8 - or, for example, a 3D reconstruction using the TB observed - similar to those for the reflectivity levels in Figure 8 - by the oxygen absorption band channels.

12) L. 324-325 and Figure 9: Certainly, the comparison between the reconstructed reflectivities and the ground-based radar observations is a very good element that makes the analysis more valid. However, it is not clear to me on which element the statement "*The reconstructed precipitation reflectivity distributions based on MWRI-RM observations are consistent with actual ground-based radar observations*" is based. A more precise analysis is needed - e.g. some statement about the areas where precipitation is detected, or about the position of the reflectivity maxima, or something similar.

**Minor Comments**

1) L. 15: "Precipitation" instead of "precipitation"

2) Figure 5: I would like to suggest that the authors add the highlighted shading to the center and right panels as well (b, c, e, f).

3) Figure 6: I would suggest that the authors add the section line to the other center panels (b, e, h, k) and a dashed line corresponding to a height of 4 km to the right panels (c, f, i, l). I suggest adding labels to the x axes (latitude values?) of the right panels (c, f, i, l).

4) Figure 7: same as Figure 6.

5) L. 273-309, Figures 6 and 7: In general, I suggest adding lat/lon references to the description of the left and center panels - e. g., L. 282: *underestimated reflectivity values in the northwestern rainbands (26 °N, 130 ° E) while overestimating those east of Khanun's center (23 ° N, 133 ° E)* - and height reference to the right panel description - e. g., L. 283 *The reconstructed melting layer structure was overly smooth (between 4 and 5 km)*. The addition of labels to the x axes of the right panels will make this analysis easier.

6) L. 298-324, Figure 8 and Figure 9: It would be useful to add a lat/lon reference linked to the name of the regions - e. g., L. 318: *over southern Beijing (-° N, -° E), central Hebei (-° N, -° E), and Tianji (-° N, -° E)* - or a label above the map.

---

## Author Comment (AC1)

Comments on the manuscript "Reconstruction of 3D precipitation measurements from FY-3G MWRI-RM imaging and sounding channels" by Yang et al.

**Overview**

The authors present a neural network approach to reconstruct radar reflectivity profiles from passive microwave observations. The network is trained using a month (October 23 to November 31, 2023) of spatially and temporally collocated radar and microwave radiometer data from the FengYun 3G (FY-3G) satellite, i.e., the 35 GHz Precipitation Measurement Radar (PMR) and the 10.65 to 190.31 GHz Micro-Wave Radiation Imager for the Rainfall Mission (MWRI-RM). The authors estimated the effects of temperature sounding channels at 50-53 and 118 GHz and polarization difference on the prediction by comparing three networks trained with different input channel combinations. The experiments show that the 50-53 and 118 GHz channels improve the reconstruction of radar reflectivity profiles, especially over land. The observation-based link between active and passive microwaves on global scales provides new opportunities to extract information from passive microwave observations as they do not rely on forward model assumptions and could be beneficial for global assimilation of passive microwaves.

**General comments**

The work is well structured, and the case studies prove the model's ability to reconstruct radar reflectivity profiles. Below are a few general comments on the neural network training and evaluation that require additional work but need to be addressed in a revised version of the manuscript.

There are various problems during the data preparation, model architecture, and training procedure that likely cause overfitting and limited generalization capabilities of all three trained models. Such a lack of generalization would make it difficult to relate their prediction skill to the information content of the model input. Despite the results being relatively close to the expectation, I would highly recommend adjusting several methodological aspects.

The authors perform a random split into training and test data sets with a ratio of 80/20. However, the input data is autocorrelated in space and time. In the current setup, a test scene can be as close as a few ten kilometers to the center of a training scene. This might lead to the model learning the reflectivity profile of single mesoscale precipitation systems rather than learning the generalized radiative TB feature's relation to reflectivity profiles. I suggest that the authors split their data into, e.g., weekly chunks. Also, the oversampling of precipitation scenes before splitting the data into training and test sets might lead to the occurrence of the same precipitation scene in training and test sets. However, this should be avoided. A way to replicate but still challenge the model would be data augmentation by rotating or flipping the TB field around the central footprint. Additionally, an independent validation set is needed to compare the three models. This was done only during case studies but without any quantification. Currently, the scores are computed only on the test data, which is problematic not only for the reasons stated above but also as its loss was used to determine the optimal parameters during early stopping. Finally, the number of model parameters exceeds the number of samples by a factor of five, although it should not exceed a factor of 0.1. The first fully connected layer contains more than 5 million parameters that would be sufficient for the model to learn a range of reflectivity profiles encountered during training.

**Response:**

Thank you for your suggestions. We agree that there are some issues with the data splitting and model architecture design. We would have liked to retrain the models according to your recommendations, but due to time constraints, we have instead chosen to test the generalization performance of the existing models using an independent sample. Specifically, we selected 288,892 samples from the period of December 1 to December 11, 2023, and preprocessed them according to the methods section. The evaluation results are consistent with previous ones, but with some differences in the details. The specific results are presented in Table 3 and Figure 3 of the revised manuscript, along with corresponding analysis updates. Additionally, to further demonstrate the model's generalization ability, we show the reconstruction performance for Typhoon Khanun, with an overall reconstruction RMSE of approximately 3.84 dBZ.

[Figure]

[Figure]

*I would like to encourage the authors to publish their prepared samples used to train, test, and evaluate the models as well as the three fully trained models to make their results reproducible.*

**Response:**

Thank you for your suggestion to make the data and trained models publicly available for reproducibility. Unfortunately, due to the large size of the sample data, we are unable to upload the full dataset. However, we have provided the code for sample processing in the manuscript, which should allow interested researchers to replicate our data processing steps. Additionally, the Level 1 products of PMR and MWRI-RM can be downloaded from the FENGYUN Satellite Data Center website (https://satellite.nsmc.org.cn).

Regarding the trained models, we are unable to provide them due to potential copyright considerations. As such, we respectfully request your understanding in this matter. We hope that the provided sample processing code and data access will still allow others to reproduce our results to a meaningful extent.

*The oversampling of precipitation scenes before splitting the data into training and test sets might lead to the occurrence of the same precipitation scene in training and test sets.*

**Response:**

Thank you for pointing this out. Upon reviewing the manuscript, we realized that our original description was inaccurate. We did not oversample non-precipitating samples as stated; instead, we downsampled them to match the number of precipitating samples. This approach was implemented to balance the dataset and ensure equal representation of both sample types during model training.

**Specific comments**

*Figure 1: This figure contains many errors. I suggest removing it entirely, especially since TRMM and GPM CO are not part of the manuscript. The MWRI-RM channels should be presented in a table, ideally linked to the three experiments following later. Correct the MWRI-RM channel definitions using Table 4 in*

*https://doi.org/10.34133/remotesensing.0097*

*Line 106: Provide a table that lists the footprint dimensions of the different channels.*

**Response:**

Thank you for your suggestions. We have removed Figure 1 as recommended and added a table detailing the specifications of MWRI-RM channels. The table is also linked to the three experiments discussed later in the manuscript, ensuring better alignment and clarity.

*Line 40: MWRI-RM observes only V-pol at 166 GHz, while GMI observes both polarizations at this frequency. How would the addition of polarization information at this frequency improve the reconstruction of radar reflectivity profiles, especially under higher scattering by snow?*

**Response:**

Previous studies analyzing polarization differences from GMI have shown that at higher frequencies such as 166 GHz, the polarimetric signals are more sensitive to the scattering effects of ice particles compared to lower-frequency channels (e.g., 89 GHz). The presence of dual polarization at this high frequency can provide valuable information about the orientation, shape, and phase of frozen hydrometeors within the precipitation system. This can be especially beneficial under intense scattering conditions induced by snow and other frozen particles, allowing the model to discriminate precipitation types better, identify layered structures (including the melting layer), and refine the retrieved vertical distributions of radar reflectivity.

However, in the current study, we do not specifically analyze the influence of polarization at the 166 GHz channel on the reconstruction of radar reflectivity profiles. While we recognize that incorporating polarization differences at this frequency could potentially enhance the accuracy and vertical structure representation—particularly in cases of heavy snow scattering—this aspect remains an avenue for future investigation.

*Line 45: How many independent height levels of liquid water content can be retrieved from dual oxygen absorption sounding channels? This information would be helpful to understand the physical limitations of reconstructing radar profiles with >100 vertical degrees of freedom.*

**Response:**

Thank you for your comment. As indicated by Han et al. (2015), each dual oxygen absorption sounding channel pair provides roughly one independent piece of vertical information in terms of liquid water content retrieval. Given that our MWRI-RM configuration includes four sets of such dual-channel pairs, we can obtain on the order of four independent vertical layers of liquid water content. This is a fundamental limitation due to the physical nature of passive microwave measurements. Unlike radar observations, which can resolve over a hundred vertical layers due to their much finer vertical resolution, microwave radiometers are inherently constrained by their weighting functions' limited vertical information content. We have added the relevant discussion in section 4 of the revised manuscript.

*Line 52: Provide a table with the accuracy for each channel. How does the channel accuracy affect the reconstructed radar profiles? Would it be helpful to inform the model on channel accuracy?*

**Response:**

Thank you for your insightful suggestion. He et al. (2024). provides a cross-comparison between MWRI-RM and GMI instruments, with the reported differences serving primarily as a reference for understanding the potential accuracy of MWRI-RM. As with any satellite-based observation, there are inherent measurement errors in both instruments, which may affect the accuracy of individual channels. However, a detailed analysis of how channel accuracy impacts the reconstructed radar profiles was beyond the scope of the current study. We agree that investigating the influence of channel accuracy is both important and necessary. For example, assessing the model's sensitivity to input uncertainties by perturbing the brightness temperatures in specific channels could help identify how robust the reconstructed radar reflectivity profiles are to observational errors. Such experiments could provide valuable insights into channel contributions and model stability. Additionally, informing the model explicitly about the accuracy or uncertainty of each channel might improve its ability to weigh the channels appropriately during training.

We acknowledge the significance of this topic and plan to address it in future work. This will include systematic experiments to quantify the impact of channel errors on the reconstruction performance and explore methods to enhance the model's robustness under input uncertainties. Thank you for raising this point, which will help guide our future investigations.

*Line 65: This sentence is unclear to me. Which direct relationships between active and passive microwave observations need to be identified? Explain why a direct relationship between TB and radar reflectivity is needed. In general, forward models can simulate both the active and passive signal. Is it due to assumptions in the forward model or due to differences in weather models and reality? This is a key question of this work that needs to be explained.*

**Response:**

Thank you for pointing out the confusion in our original wording. We have removed the problematic sentence in the revised manuscript. Our intention was not to overstate the direct correlation between active and passive microwave observations but rather to emphasize that meaningful relationships exist and can be explored using deep learning techniques. By doing so, we aim to better understand how two fundamentally different types of observations—active radar data and passive microwave radiances—can be mapped onto each other to retrieve more detailed three-dimensional precipitation structures.

*Line 73: It would be helpful to have a table of previous work and the respective methods. How does the method used in this work differ from the Res-UNet in Brüning et al. (2024), and how are the differences motivated? What is the accuracy of other work that replicates radar reflectivity profiles?*

**Response:**

Thank you for the suggestion. In the revised manuscript, we have reorganized the literature review to focus on a selection of representative studies that are most relevant to our work. Due to time constraints, we have not provided a comprehensive table listing all related studies. Instead, we have highlighted some key works that illustrate the range of methodologies, data sources, and achieved accuracies in **Lines 69-77**. This approach allows us to give a more detailed and meaningful comparison without overwhelming the reader.

Our study primarily builds upon the work of Yang et al. (2024), which pioneered the use of passive microwave (PMW) radiances from GMI to reconstruct DPR reflectivities. While their study achieved an RMSE below 4 dBZ across all altitudes by incorporating polarization differences and temperature profiles, it was limited to oceanic precipitation scenarios. In contrast, we extend their approach by leveraging the newly launched FY-3G satellite's MWRI-RM and PMR instruments. The addition of dual oxygen absorption channels (50 and 118 GHz) addresses the challenges of reconstructing reflectivity over land and provides more detailed information about the melting layer. We also explicitly analyze the impact of polarization differences, a factor not thoroughly examined in Yang et al. (2024), to further refine our retrievals and capture precipitation structures with greater fidelity.

*Line 76: What is meant by "limited to specific precipitation scenarios due to limited spectral channels of GMI"? Are only specific scenarios used to train the model in that work?*

Response:

Thank you for your comment. The statement refers to the fact that the previous work focused on reconstructing radar reflectivity specifically for oceanic precipitation scenarios. This limitation arises because the GMI sensor has fewer spectral channels, which restricts its ability to capture the diverse atmospheric and surface conditions necessary for reconstructing reflectivity over land. We have clarified this point in the revised manuscript in **Lines 82-83**.

*Line 85: It might be confusing to mention 3D structures when the output of your model is 1D. The network does not know about the 3D nature of radar reflectivity fields and is not forced to be spatially consistent in the radar reflectivity space. Only the consecutive application of the model along and across track dimensions leads to a 3D field. Would an improved model performance be expected if the output is a 3D field? Why was this approach not chosen for this work?*

**Response**:

Thank you for your insightful comment. We agree that the vertical information provided by passive microwave observations has certain limitations, making it challenging to fully achieve realistic 3D radar reflectivity reconstruction. In this work, we focused on using 2D multi-channel brightness temperature data to reconstruct multi-layer reflectivity profiles. This approach was chosen to leverage spatial information in the brightness temperature data to

compensate for the limited vertical information.

In future work, we aim to address this challenge by incorporating more diverse input data sources and adopting more powerful deep learning architectures. These enhancements could enable more complete 3D radar reflectivity reconstructions, improving spatial consistency and capturing the 3D nature of radar reflectivity fields.

*Line 86: Why are the non-precipitating scenes not split into land and ocean as well?*

**Response**:
Thank you for your comment. We have now split the non-precipitating scenes into land and ocean categories, and the corresponding evaluation results are presented in Table 3.

**Figure 2b: Mention how the footprints are scaled in size and number instead of adding "not to scale." What does "*Npoints" mean? Indicate the footprints used for training here and combine them with Fig. 3.**
**Figure 3: This should be added to Fig. 2 for better understanding. Indicate the size of a 15x15 patch in kilometers.**

**Response:**
Thank you for your helpful suggestions. "Npoints" refers to the number of scanning points along each scan line, and we have added an explanation of this in the revised manuscript in the caption of Figure 1. Figure 2b is a schematic illustration of the observation geometry of PMR and MWRI-RM, and it is not intended to represent an exact scale based on real observations. The precise instantaneous field of view for each channel is provided in Table 1. The footprints used for training are shown in Figure 3, which correspond to 15x15 footprints around each matching point. We have merged Figures 2 and 3 for better clarity and understanding, as suggested.

*Line 115: Are the Level 1 products of PMR corrected for two-way attenuation by liquid water and water vapor? If not, how does this affect the reconstruction and comparison with PMR observations under varying incidence angles? How does multiple scattering affect the reconstruction in the presence of high-density ice particles typical for extreme precipitation events?*

**Response**:
Thank you for your comment. The PMR Level 1 product we used has not been corrected for two-way attenuation by liquid water and water vapor. We acknowledge that varying incidence angles result in different path lengths, leading to varying degrees of attenuation. However, in this study, we only used reflectivity profiles with an incidence angle less than 2° for training, where the differences in path length are negligible. For consistency, our model evaluations also focused primarily on reconstructing reflectivity profiles with an incidence angle below 2°. As the model directly learns relationships from the data, it is possible that it has implicitly accounted for some attenuation effects caused by precipitation particles. This remains an interesting topic for further investigation. The PMR Level 2 product provides reflectivity

corrected for attenuation, and we plan to use this product in future work to refine our reconstructions and analyze the differences between using corrected and uncorrected reflectivity data. This will help better understand the effects of attenuation and multiple scattering under extreme precipitation conditions.

*Line 116: Why are case studies using data from July 2023 if the data is released after October 23, 2023? Clarify this. I assume the model was trained on the October/November data. If that is true, do you expect the same performance for other months of the year? What is meant by "used here for the preliminary research"?*
*Line 259: The month of July is not mentioned in the data section.*

**Response**:
Thank you for your comments. Before the official data release, we were granted access to PMR observations from July 2023, including data from Typhoon Khanun and the Beijing extreme rainfall event. These two cases represent extreme precipitation events, making them valuable for further evaluating the model's generalization ability. We have clarified this in the revised manuscript in **Lines 281-283.**

Our model was trained using one month of officially released data from October 23 to November 31, 2023. Due to the limited amount of officially released data available at the time of our research, we used only one month of data for training. We acknowledge that this may affect the model's generalization ability. However, the trained model still demonstrated reasonably good reconstruction performance when applied to independent test samples, including these July case studies.

In future work, we plan to use more data to fine-tune the current model and further enhance its generalization ability. We have clarified this information in the revised manuscript, including the mention of July in the data section. Thank you for highlighting these points.

*Line 126: Mention the PMR scan interval of 0.7° and provide the number of cross-track scans that match the 2° incidence angle criterion.*

**Response**:
Thank you for your comment. We have included the PMR scan interval of 0.7° in the revised manuscript in **Line 104.** Based on the 2° incidence angle criterion, five cross-track scan points per scan line meet this requirement. This information has been included in the revised manuscript in **Line 136.**

*Line 130: A threshold of 8 km cuts off the reflectivity profiles during the case studies. Also, the additional 183.31+/-2 GHz channel of MWRI-RM compared to GMI peaks around this height, depending on humidity levels. By how much does the threshold of 8 km change the ratio of clear-sky and cloudy pixels compared to a 9 or 10 km threshold?*

**Response**:
Thank you for your comment. Based on our analysis of reflectivity profile samples under

precipitation conditions, the ratio of clear-sky to cloudy pixels is approximately 0.4 when using an 8 km threshold. When the threshold is increased to 10 km, this ratio decreases to approximately 0.3.

*Line 133: How do 136 range bins from 1.1 to 8 km match with the range resolution of 250 m of PMR? Also, note that in Table 1, the output is 138. Why are both different? How was the noise, sidelobe clutter, and ground clutter filtered, and is the filter considered surface type?*

**Response**:

Thank you for your comment. The mention of 136 range bins in Line 133 is incorrect and has been revised to 138. The PMR samples in the range direction at 50 m intervals, resulting in 138 bins from 1.1 to 8 km.

The noise, sidelobe clutter, and ground clutter were filtered based on the quality flag data (flagEcho) provided in the PMR's HDF file. This ensures the filtering process considers the surface type and other relevant quality indicators.

We appreciate your feedback and have corrected this in the revised manuscript.

*Line 138: What does NaN mean in the context of the loss function? How is terrain treated, i.e., subsurface PMR pixels? It would be important information for the reconstruction to know the lowest boundary of the precipitation field. In Fig. 7, it seems that mountains extend up to 2 km, and the models add precipitation below the surface (see. Fig. 7i right above the letter A)*

*Line 236: How much of the uncertainty over land can be attributed to the lack of topography information?*

**Response**:

In the context of the loss function, NaN indicates bins that are marked as invalid due to the influence of sidelobes or ground clutter, which can negatively affect the reconstruction. These bins are excluded during loss calculation to ensure that they do not introduce bias or degrade the model's performance. By skipping these NaN values, the loss function focuses only on valid data points, leading to a more robust and accurate reconstruction.

Thank you for your insightful comments. We have indeed noticed the issue of reflectivity appearing below the surface when reconstructing reflectivity profiles over land. In our future work, we will address this problem by developing a targeted model specifically for reconstructing land-based precipitation reflectivity.

We plan to incorporate the strategies, such as adding terrain height information as an input feature, applying post-processing techniques to mask invalid reflectivity values below the surface, and designing the loss function to account for surface constraints. These improvements aim to enhance the model's accuracy and ensure physically consistent outputs. The discussion on this issue is provided in the revised manuscript in **Lines 348-354**.

*Line 143: Could it happen that two PMR columns are assigned to the same MWRI-RM footprint due to their different spatial resolution? Are those duplicates filtered out?*

**Response**:

Thank you for your comment. In our matching process, the MWRI-RM observation point is used as the target, and each MWRI-RM point is matched to the nearest PMR observation point. This ensures that each MWRI-RM footprint corresponds to a single PMR column, avoiding any duplication issues.

We have clarified this process in the revised manuscript in **Lines 148-150** to address your concern. Thank you for bringing this up.

*Line 152: How is a precipitation event defined? Provide information on the filter method or threshold.*

*Line 215: See comment on line 152.*

**Response:**

Thank you for your comment. In our analysis, precipitation events were defined based on the precipitation flag (flagPrecip) provided in the PMR data. This flag distinguishes between precipitating and non-precipitating samples.

We have clarified this definition and the use of flagPrecip in the revised manuscript in **Lines 164-165** to ensure transparency and address your concern. Thank you for pointing this out.

*Line 155: How many samples were available prior to the oversampling? Based on this description and the diagram in Fig. 4, it seems that oversampling was performed prior to the data splitting. Clarify this. See also the general comment regarding random data splitting.*

**Response**:

Thank you for pointing this out. Upon reviewing the manuscript, we realized that our original description was inaccurate. We did not oversample non-precipitating samples as stated; instead, we undersampled them to match the number of precipitating samples. This approach was implemented to balance the dataset and ensure equal representation of both sample types during model training.

Regarding the number of samples prior to undersampling, while we did not explicitly record the exact number of non-precipitating samples, it is clear that this number was significantly larger than the undersampled total, as non-precipitating samples far outnumbered precipitating ones in the original dataset. We will clarify this in the revised manuscript in **Lines 162-164**.

*Line 157: Describe the standardization method.*

**Respons:**

Thank you for your comment. The standardization method used in our study follows the approach shown in Equation (1):

$$x' = \frac{x - \mu}{\sigma}$$

where x is the original value, μ is the arithmetic mean, and σ is the standard deviation. This normalization is applied independently to each channel using the mean and standard deviation.

This information has been included in the revised manuscript in **Lines 172-175**.

*Line 157: Does logarithmic transformation mean Ze was transformed to dBZ? Clarify this.*

**Response:**

The reflectivity data provided in PMR Level 1 is already in **dBZ**. In our preprocessing, we applied a logarithmic transformation to these dBZ values using np.log(). This transformation was not a unit conversion, but rather a preprocessing step aimed at enhancing the model's learning process. The logarithmic transformation was used to normalize the data distribution and reduce the range of values, which helps the model better handle variations in the reflectivity data

**Figure 4: Explain the meaning of "values adjustment" in the PMR branch.**

**Response:**

Thank you for pointing out the ambiguity in the term "values adjustment" in the PMR branch. We agree that the original description was unclear, and we have updated it in the revised manuscript for better clarity.

Specifically, "values adjustment" refers to **noise marking** in the PMR branch. Radar bins affected by noise were assigned a baseline value of 10 dBZ. This baseline value helps to differentiate non-precipitating and precipitating conditions while minimizing the influence of radar noise on the reconstruction process. Importantly, this adjustment does not significantly alter the overall distribution of reflectivity values, ensuring the integrity of the reconstructed profiles.

*Table 1: The number of parameters is very large and exceeds the number of samples. The first dense layer contains 28800 * 200 + 200 (5.76 M parameters). This is much more than the number of samples (about 1 M) and will likely lead to poor generalization of all three trained models. Pooling layers could help to reduce dimensionality before passing to dense layers. Also, note that the number of parameters is not correctly shown in the table because the bias terms are missing for Conv (32, 64, 128) and FC (200, 138). It might be more meaningful to show the activation shape, activation size, and total number of parameters for each layer. Mention the kernel size in the text. Also, see the comment on line 133 regarding the confusion about the vertical resolution of the Ze profile (136 or 138 bins).*

**Response:**

Thank you for your detailed feedback and suggestions. We acknowledge the concerns

regarding the large number of parameters, particularly in the fully connected layers, and the potential for overfitting due to the current architecture. While we agree that introducing pooling layers could help reduce dimensionality and improve generalization, due to time constraints, we are unable to modify the model architecture and retrain the models for this version of the manuscript.

To address your comments, we have revised Table 2 to provide accurate information, including the bias terms for the convolutional and fully connected layers. Additionally, we now include the activation shapes, activation sizes, and the total number of parameters for each layer to ensure clarity. The kernel size for convolutional layers (3×3) has been explicitly stated, and the table reflects the current model architecture used in our experiments.

We acknowledge that the parameter count in the dense layers is large relative to the sample size, which may limit the model's generalization ability. This is an important observation, and we will explore architectural refinements such as the addition of pooling layers or regularization techniques in future work to optimize model performance and efficiency.

Thank you for raising these valuable points, which will guide our future improvements. For now, the updated table ensures transparency regarding the model's current configuration.

*Line 170: Use shorter names for the experiments (e.g., Ex14, Ex26, etc.).*

**Response:**

Thank you for your suggestion. We have simplified the experiment names to shorter forms, such as Ex14, Ex26, etc., throughout the manuscript to improve clarity and readability. We appreciate your feedback, which has helped enhance the presentation of our work.

*Line 171: Mention the batch size, number of epochs / early stopping, learning rate, and other details on training (restore best weights, etc.). Add figures of the training and test loss for each epoch until training is stopped for each of the three models to the appendix.*

**Response:**

Thank you for your comment. We have included the requested training details in the revised manuscript. Specifically, we used eight A800 GPUs to train the models with a batch size of 512 and 100 epochs. The learning rate was initialized at 1e-3 and adjusted using an InverseTimeDecay schedule. Early stopping was applied, and the best weights were restored during training. This information has been included in the revised manuscript in **Lines 201-203**.

Additionally, we have added the loss curves for training and testing across epochs for each of the three models in the appendix. Thank you for your valuable suggestion, which has helped improve the clarity and transparency of our methodology.

*Line 173: Clarify why the published code has a different loss than MSE (sum of squared errors: tf.reduce_sum(tf.square(y_true_filtered - y_pred_filtered))) in line 58 of training_cnn.py.*

Thank you for pointing out the discrepancy regarding the loss function used in our published

code versus what is stated in the manuscript. Upon review, we realize that our manuscript's description may have caused some confusion. To clarify:

In **line 58 of training_cnn.py**, the actual loss function implemented is **Sum of Squared Errors (SSE)**, defined as:

tf.reduce_sum(tf.square(y_true_filtered - y_pred_filtered))

This is consistent with the published code. However, in the manuscript, we referred to the loss as "MSE," which was imprecise. We appreciate your feedback, and we will revise the manuscript to explicitly state that **SSE** was used as the loss function.

The reason for choosing SSE over MSE is based on findings from our previous studies. In the context of radar reflectivity reconstruction, the following characteristics of the task make SSE more appropriate:

1. **Emphasis on Overall Reconstruction Error**: Reflectivity reconstruction involves capturing the full profile's structure across all vertical levels. SSE directly sums up the reconstruction error across all samples and levels without normalization, ensuring that significant deviations in key regions are not diluted.

2. **Sensitivity to Sparse High Values**: Reflectivity profiles often feature sparse but crucial high values (e.g., during precipitation events) embedded in a majority of background values. SSE's quadratic nature amplifies the contribution of these high-value regions, leading the model to focus on reconstructing these physically meaningful regions more accurately.

3. **Handling of Imbalanced Vertical Distributions**: Reflectivity values tend to exhibit significant vertical variability, with lower levels often containing larger and more impactful values for precipitation and hydrometeorology. SSE effectively accounts for these variations by prioritizing larger deviations without introducing explicit weighting mechanisms.

We have clarified in the revised manuscript in **Lines 203-210** to clarify the above rationale and explicitly state that the loss used in our code and experiments is **SSE**.

*Line 174: It would be good to separate the description of experiments from the model architecture chapter. The paragraphs of each experiment repeat large parts of the introduction and do not belong to the methodology section.*

**Response:**

Thank you for your suggestion. We agree that the detailed explanations of the three experiments were somewhat redundant in the methodology section. In the revised manuscript, we have streamlined this section by retaining only the descriptions of the different inputs for each experiment. This ensures clarity and avoids unnecessary repetition while maintaining focus on the methodology.

We appreciate your feedback, which has helped improve the structure and readability of the manuscript.

*Line 193: Explain how the model output advances weather forecasting techniques and links to hydrometeors simulated by weather models.*
*Line 225: See comment on line 193.*

**Response:**

Thank you for your comment. Upon review, we recognize that the original statement regarding the advancement of weather forecasting techniques was unclear and, at this stage, speculative. As the operational applicability of the reconstructed reflectivity profiles remains to be further evaluated, we have removed the related discussion from the manuscript to avoid overstating the findings.

*Line 231: It would be good to split non-precipitation in land and ocean as well; see comment on line 86. How does the model perform in coastal regions where parts of the passive channel are affected by land?*

**Response:**

Thank you for your suggestion. We agree that splitting non-precipitation cases into land and ocean categories, as well as analyzing coastal regions where passive channels are influenced by land, provides valuable insights into the model's performance. We have filtered these samples and included them in the evaluation. The results have been added to Table 3 in the revised manuscript, along with the corresponding analysis.
We appreciate your feedback, which has significantly improved the thoroughness of our evaluation.

*Table 2: Reduce the precision of metrics to those digits that are significant. The names of the experiments are wrong; all are named "baseline." The F1 score is the same for land and ocean for all three models. Is that correct?*
*Line 254: How does the F1 score vary between land and ocean? See comment on Table 2.*

**Response:**

Thank you for your comment. In the revised manuscript, we have addressed the concerns regarding Table 2. We have corrected the experiment names, reduced the precision of metrics to reflect only significant digits, and clarified the results. Additionally, we have decided to exclude the F1 score from the analysis to avoid potential confusion, as it did not provide meaningful distinctions between land and ocean cases.

*Line 234: In general, the melting layer is below 3 km in mid-latitudes, which are also covered by the satellite. Why does the RMSE show a peak at 4-5 km only and not at lower heights?*

**Response**:

Thank you for your comment. Currently, we do not have access to PMR-specific melting layer products. Therefore, we referred to the analysis presented in the work by Hu et al. (2024), which investigates the quasi-global climatological features of the melting layer over the latitude range of 65°S to 65°N by Utilizing the detection from the Dual-frequency Precipitation Radar onboard the Global Precipitation Measurement Mission Core Observatory

during 2018-2022.

Their study shows that a larger amount of ML occurs more frequently in tropical regions (30°S–30°N), especially in the Asia-Pacific region (fig. 1d). The melting layer height varies with latitude, being higher in tropical regions (30°S–30°N) and lower in mid- and high-latitudes (fig. 1a and d). Specifically, in tropical areas (0°–30°N), the top of the melting layer is generally distributed between 4–5 km, and the bottom height is primarily between 2–4 km. This finding aligns with our results.

In the revised version of the manuscript (**Line 243**), we have incorporated the referenced work by Hu et al. (2024). By including this reference, we aim to make our analysis more robust and provide a clearer context for the RMSE peak at 4–5 km.

[Figure]

**Fig. 1** Global geographical distributions of mean melting layer top height (**a**), melting layer geometric thickness (**b**), melting layer bottom height (**c**), and melting layer amount (**d**) detected by GPM DPR from 2018 to 2022 (plotted at 1° ×1° grid resolution)

**Fig. 6** The monthly mean of the top height of the melting layer detected by GPM DPR in five terrain areas (elevation ~0, 0~0.2, 0.2~0.5, 0.5~1, and 1~ km) at four typical latitudinal regions: (**a**) 0°~30°N, (**b**) 30°N~65°N, (**c**) 0°~30°S, (**d**) 30°S ~65°S

Hu, X., Ai, W., Qiao, J., and Yan, W.: Insight into global climatology of melting layer: latitudinal dependence and orographic relief, Theor Appl Climatol, 155, 4863–4873, https://doi.org/10.1007/s00704-024-04926-6, 2024.

*Line 244: The scattering signal of precipitation at the 50-53 and 118 GHz channels is very small.*

**Response:**

Thank you for your feedback. We have removed the phrase "the scattering signal of precipitation" in the revised manuscript to address this issue and avoid potential misunderstandings. We appreciate your careful review and constructive suggestions.

*Line 260: Explain why these events are "challenging" for the model. Those are strong precipitation events with clear passive microwave signatures.*

**Response:**

Thank you for your comment. While strong precipitation events often exhibit clear passive microwave signatures, they remain challenging for deep learning models due to their extreme nature. These events typically involve highly complex and dynamic atmospheric conditions, which can introduce significant variability and non-linear interactions that are difficult for the model to generalize. Additionally, extreme precipitation often pushes the model beyond the range of conditions it has encountered during training, further impacting its performance.
We will clarify this in the revised manuscript in **Lines 278-281** to provide a more detailed explanation of the challenges associated with these events.

*Line 280: The discussion following this line is very vague and subjective. I recommend quantifying the comparison between PMR and the reconstructed profiles and providing a difference and scatter plot between the observed profile and the predictions.*
*Line 303: Provide a quantitative comparison between observation and prediction.*

**Response:**

Thank you for your suggestion. We have incorporated two quantitative metrics (RMSE and MBE as) and scatter plot to evaluate the reconstruction performance of different models for the specific case study.

*The discussion on the melting layer could be supported by comparing it with reanalysis data.*

**Response:**

We appreciate your suggestion to incorporate reanalysis data into our discussion of the melting layer. However, our primary objective was to assess the model's capability to reconstruct the vertical structure of reflectivity, particularly the position of high-reflectivity features such as the bright band. The bright band serves as an indirect indicator of the melting layer, and its accurate representation in our modeling framework already implies a reasonable portrayal of the underlying thermodynamic structure. This correspondence between the observed and reconstructed reflectivity profiles allows us to infer that the model has captured the melting layer's position effectively without explicitly computing it. Direct comparison with reanalysis data, such as the zero-degree-level fields, falls outside the scope of our current analysis. Nevertheless, we acknowledge the value of such comparisons and will consider them in future studies to further validate and refine the model's performance.

*Figure 5: Use the same x-axis among the columns to make them comparable, ideally starting at 0 for RMSE and STD. How was the yellow area in a and d calculated, considering that the melting layer varies with latitude? In panels g and I, the polarization model performs worse than the model without polarization close to the surface. Why?*

**Response:**

Thank you for your suggestions. We have revised Figure 5 to use a consistent x-axis across all columns, starting at 0 for RMSE and STD, to facilitate comparison. Regarding the yellow shaded area in panels a and d, it represented the empirically estimated range of melting layer heights. However, as it was not based on a specific calculation and could potentially lead to misunderstanding, we have removed this shaded area in the revised figure to avoid confusion.

*Line 294: What is meant by "vertical resolution," and how was it determined?*
*Line 354: Clarify if resolution means vertical or horizontal.*

**Response**:

Thank you for pointing this out. We agree that the term "vertical resolution" may be misleading. While the reconstructed reflectivity profiles have the same resolution as the actual observations in terms of their vertical discretization, the inherent ambiguity of the passive microwave signals used as input affects the final output accuracy. This can lead to limitations in capturing finer details, such as the exact height of high-reflectivity regions like the melting layer.

We have revised the manuscript to avoid the use of "vertical resolution" and provide a more precise explanation of the factors influencing the reconstruction accuracy. Thank you for highlighting this important clarification.

*Line 295: Add a reference to the precipitation amount.*

**Response:**

Thank you for your comment. We have revised the text to include a detailed description of the precipitation amount during the "23·7" BTH extreme rainfall (BTHER) event, based on the findings of Zhao et al. (2024).

Zhao, D., Xu, H., Li, Y., Yu, Y., Duan, Y., Xu, X., and Chen, L.: Locally opposite responses of the 2023 Beijing–Tianjin–Hebei extreme rainfall event to global anthropogenic warming, npj Clim Atmos Sci, 7, 1–8, https://doi.org/10.1038/s41612-024-00584-7, 2024.

*Line 296: Mark the regions mentioned in the text inside the map.*

**Response**:

Thank you for your suggestion. We have added the location markers for the regions mentioned in the text to the map in the revised version of the manuscript to enhance clarity and alignment between the text and figures.

*Line 321: Add ground-based radar data to the data section.*

**Response**:

Thank you for your suggestion. We have added a description of the ground-based radar data in the data section in **Lines 115-121**. Specifically, we utilized the dual-polarization radar from

the China Next Generation Weather Radar (CINRAD/SA) network for comparison with the land precipitation reflectivity reconstruction results. The CINRAD/SA products have a radial distance resolution of 250 m and an azimuthal resolution of 1 degree. The Volume Coverage Pattern 21 (VCP21) scan mode was selected, which sweeps 9 elevation angles of 0.5, 1.5, 2.4, 3.4, 4.3, 6.0, 9.9, 14.6, and 19.5 degrees in 6 minutes (Teng et al., 2023). Based on the interpolation method used in Xiao and Liu (2006), we transformed the radar data from spherical coordinates into a unified Cartesian coordinate system, creating a 3D grid. The radar data were then stitched and organized to generate the CAPPIs (Constant Altitude Plan Position Indicator) data used in this study.

*Line 321: What does "consistent" mean? It is not obvious from Fig. 9 and Fig. 8b. Plot both data in one figure with the same geographic extent and color bar. Ideally, a simple scatter plot should be made to show that the model and observation agree.*

**Response**:

Thank you for your suggestion. In response to your comment, we have merged the two figures into one, ensuring that they share the same geographic extent and color bar for direct visual comparison. Additionally, we have revised the manuscript to provide a more precise analysis of the reconstructed reflectivities and their comparison with ground-based radar observations (**Lines 370-374**). This includes specific statements about areas where precipitation is detected and the position of reflectivity maxima to make the comparison more explicit.

We agree that a quantitative comparison, such as a scatter plot, between the ground-based radar reflectivity and the reconstructed spaceborne radar reflectivity would offer valuable insights. However, due to differences in radar wavelength, spatial resolution, and scanning geometry, such a quantitative comparison requires significant additional effort. This includes aligning the datasets, compensating for different measurement characteristics, and addressing inherent discrepancies.

Given the scope and time constraints of this work, we are unable to provide this analysis in the current manuscript. However, we plan to address this in future work by focusing on a more detailed quantitative evaluation of the reconstructed reflectivity profiles, particularly in regions beyond the PMR's actual scanning area, to further validate the model's performance.

*Line 353: What is meant by representative training dataset? Why is the data used here not representative?*

**Response**:

At line 353, we initially referred to having a "representative training dataset," which may have been misleading. The data used in this study were collected within a one-month period (2023-10-23 to 2023-11-31), and, as a result, the dataset did not fully capture the variability and diversity of global precipitation scenarios, particularly for land-based precipitation in the Northern Hemisphere. This limited data coverage reduces the model's ability to generalize well to a wide range of conditions, as it has fewer samples from certain environments (e.g., less land precipitation data from the Northern Hemisphere). To clarify this point, we have

removed the term "representative" from the revised manuscript. In future work, we plan to incorporate a more diverse and extensive dataset spanning multiple seasons, regions, and precipitation types, which will help improve the model's generalization capabilities.

*Line 363: Mention snow scattering as well.*

**Response**:
Thank you for your suggestion. We have included a mention of snow scattering in the revised manuscript as requested.

*Line 369: What exactly is the advantage of GANs and diffusion models compared to the method selected here? This needs more explanation, potentially linked with the introduction where previous work is presented.*

**Response**:
Thank you for your suggestion. The generative models have demonstrated remarkable capabilities in capturing fine-grained details and generating realistic outputs in various fields, which makes them relevant for radar reflectivity reconstruction as well.

GANs are particularly known for their ability to model complex data distributions and produce high-quality, detailed reconstructions. As shown in the work of Leinonen et al. (2019), a CGAN framework was applied to generate two-dimensional cloud vertical structures that would be observed by the CloudSat satellite-based radar, using only four input variables derived from MODIS (cloud top pressure, cloud optical depth, effective radius, and cloud water path). The CGAN in their study was capable of generating sharp and realistic images that closely resembled radar reflectivity fields. Furthermore, it demonstrated robustness to missing input data by effectively interpolating into regions with data gaps and was able to exploit spatial structures in the input data to infer features such as multilayer clouds. This example highlights how GANs, particularly CGANs, can leverage their generative capacity to recover subtle spatial variations and infer complex structures in atmospheric systems.

Diffusion models, on the other hand, provide an alternative generative modeling framework that excels in generating high-resolution and diverse samples. These models gradually transform noise into a structured data distribution through an iterative denoising process, allowing them to capture fine-scale spatial details and subtle variations that are often challenging for traditional deterministic methods. Their ability to preserve high-resolution information during generation makes them particularly suitable for reconstructing spatially complex systems, such as precipitation structures observed in radar reflectivity profiles.

While our current method adopts a deterministic approach to reconstruct radar reflectivity profiles, which ensures computational efficiency and straightforward implementation, it may face challenges in capturing finer-scale features or handling under-sampled regions. By contrast, generative models such as GANs and diffusion models have the potential to address these limitations by providing more nuanced reconstructions and effectively dealing with missing or noisy input data. In the context of PMR reconstruction, leveraging these models could improve the quality and resolution of the reconstructed reflectivity profiles, particularly in cases involving complex precipitation systems or incomplete data coverage.

In the revised manuscript, we have referred to the works of Leinonen et al. (2019) to make our discussion more profound **(Lines 425-430).**

*Line 372: "fully replicate" sounds as if it is, in principle, possible to create a radar reflectivity profile from passive microwave observations equivalent to a radar observation with the right AI method. I suggest to rewrite this. It is impossible to retrieve unique information on each height bin from the limited information of passive microwave observations.*

**Response**:

Thank you for pointing this out. We agree that our original wording may have been too absolute. It is indeed not feasible to fully replicate radar reflectivity profiles from passive microwave observations due to the inherent limitations in the vertical resolution and information content of passive microwave data. We have revised the statement to reflect this limitation more accurately and acknowledge that while AI methods can reconstruct radar-like profiles with useful detail, they cannot fully replace the unique vertical information provided by active radar observations. We have clarified in the revised manuscript in **Lines 430-434**.

*However, it would be interesting to see which channels provide information on the height level of the reflectivity profile. Could this be seen when computing the Jacobian of the model with 35 input channels?*

**Response:**

Thank you for your suggestion. We agree that analyzing the contribution of each channel to the height levels of the reflectivity profile, for instance by computing the Jacobian of the model with respect to the 35 input channels, is a valuable approach. Such an analysis can indeed reveal which channels provide the most information regarding the vertical structure of the reflectivity.

We are currently conducting related work to extract gradient-based information to assess the influence of input channels on the output reflectivity profiles. However, due to time constraints, we were unable to include this analysis in the present study. We plan to incorporate this analysis in future work, as it can provide deeper insights into the role of individual channels and further improve the interpretability and performance of the model.

Thank you again for your valuable suggestion, which will help guide our ongoing and future research.

**Technical corrections**

Line 7: What does "VPR" stand for? I suggest using "reflectivity profiles." and not "VPR."

Line 15: The start of the sentence should be a capitalized letter.

Line 22: Mentioning nations is distracting. Remove them throughout the manuscript.

Line 25: See comment on line 22.

Line 28: Remove the word "wideband."

Line 32: Rephrase the beginning of the sentence to more neutral wording like "To further

advance global precipitation observations."

Line 49: Remove the word "strong."

Line 56: The year in the reference is missing.

Line 61: See comment on line 18.

Line 73: Does "precipitation measurement radar" refer to PMR or any radar?

Line 90: The "HIW" acronym is not needed.

Line 93: See comment on line 22.

Line 96: DPR is written instead of PMR.

Line 104: Replace "frequency points" with "channels."

Line 112: Move this to the data availability section.

Line 128: Remove the word "detrimental."

Line 165: Remove the word "advanced."

Line 168: Remove the word "enriched."

Line 249: Replace "precipitation reflectivity" with "radar reflectivity."

Line 294: Replace the word "fuzziness."

**Response:Thank you for your careful scrutiny, we have revised it accordingly!**

**Line 18: The acronym "AWR" is rarely used. Use "radar" instead.**

**Response:**

Thank you for your suggestion. While "radar" is indeed commonly used, we adopted the term "Active Microwave Sensors (AMW)" following its usage in recent literature, such as in the works of de Roda Husman et al. (2021) and Sharifnezhadazizi (2022). These references employ the term to emphasize the distinction between active and passive microwave sensors in remote sensing applications. However, we understand that clarity and common terminology are crucial for broader readership. We will revise the manuscript to use "radar" instead, ensuring consistency and accessibility for the audience.

de Roda Husman, S., Lhermitte, S., and Wouters, B.: Towards Improved Spatio-Temporal Resolution Surface Meltwater Detection on the Antarctic Ice Shelves from the Synergy of Active and Passive Microwave Remote Sensing, 2021, C52B-04, 2021.

Sharifnezhadazizi, Zahra, "Data Fusion and Synergy of Active and Passive Remote Sensing; An application for Freeze Thaw Detections" (2022). *CUNY Academic Works*.

---

## Author Comment (AC2)

*What is the major difference of reconstruction method between this paper and your previous study (Yang, Y., Han, W., Sun, H., Xie, H., and Gao, Z.: Reconstruction of 3D DPR Observations Using GMI Radiances, Geophysical Research Letters, 51, e2023GL106 846, https://doi.org/10.1029/2023GL106846, 2024.), Except for the data, it seems that the methods used in the two studies are similar.*

**Response:**

Thank you for your question. While the models used in both studies are similar, the key difference lies in the architecture and the data incorporated. In our previous study, we employed a hybrid network architecture to incorporate temperature profiles, which included both a CNN module and an MLP module. The CNN module was used to extract spatial features from the multi-channel GMI brightness temperatures, while the MLP module was designed to extract temperature profile information from ERA5. Through evaluation, we found that incorporating temperature profiles significantly improved the model's reconstruction accuracy, particularly near the melting layer.

In contrast, in this study, we use additional channels from the MWRI-RM, including multiple oxygen absorption channels (50 GHz and 118 GHz). These channels are sensitive to both clouds and precipitation, and also provide atmospheric vertical temperature information. Therefore, the model architecture in this study is based solely on a CNN. A detailed explanation of this difference can be found in the revised manuscript, **lines 177-182**.

*What is the spatial resolution of the reconstructed reflectivity profiles?*

**Response:**

Thank you for your question. It is challenging to define an exact spatial resolution for the reconstructed reflectivity profiles, as this depends on the model's reconstruction accuracy. Our goal is to ensure that the resolution of the reconstructed reflectivity profiles aligns as closely as possible with the resolution of PMR.

*What exactly are the channels used in each experiment? I think it would be better to show a list of channels used in each experiment.*

Response:

Thank you for your valuable suggestion. In response, we have added a table in the manuscript detailing the specifications of the MWRI-RM channels, which will help readers better understand the channel information. Additionally, we have clearly indicated the number of oxygen absorption channels and polarization difference sets in the manuscript. To further clarify the input channels for the three experiments, we have provided the following detailed explanation:

**Line 107-114**:

"MWRI-RM: A significant payload of the FY-3G satellite, the MWRI-RM has 17 frequency points ranging from 10.65 GHz to 183 GHz, including nine dual-polarized channels in the 10-89 GHz spectrum and twelve oxygen absorption sounding channels around 50 GHz and 118 GHz, totaling 26 channels. These channels have spatial resolutions ranging from 5 to 25 kilometers. Detailed channel information for the MWRI-RM can be found in Table 1

Using a conical scanning regime with imaging channels and sounding channels having incidence angles of 53°±1° and 50°±1°, respectively, MWRI-RM has an effective observation swath of 800 kilometers, as shown in Fig. 1b. This instrument captures passive microwave radiation from the Earth's surface, providing valuable information on precipitation, atmospheric water vapor, cloud liquid content, path-integrated liquid water thickness, melting layer height and thickness, sea surface wind speed, and more (Zhang et al., 2023a)."

**Line 186-195**:

"To critically evaluate the influence of various channel configurations and feature inputs on the model's ability to reconstruct radar reflectivity, we orchestrated a series of controlled experiments, as shown in Fig. 2. The baseline experiment (Ex14) excludes the oxygen absorption channels, using 14 input channels (out of the total 26, excluding 12 oxygen absorption channels) to assess the model's basic capability without temperature information. The full channel experiment (Ex26) incorporates all 26 input channels, including the oxygen absorption bands, providing additional temperature profiles to improve precipitation detection and enhance the delineation of melting layers. Building on this, the polarization difference enhanced experiment (Ex35) adds Tb polarization difference data, bringing the total number of input channels to 35 (26 channels plus 9 polarization difference sets), which enhances the model's ability to distinguish precipitation types and capture finer structural details. With a consistent architecture across all experiments, this setup allows for a rigorous evaluation of how the extended MWRI-RM channel information influences the model's performance in precipitation reconstruction."

*In "Full Channel Experiment", all 26 channels of MWRI-RM were used in the training model, and in "Polarization Difference Enhanced Experiment", 9 additional Tb polarization difference data were added in the training model, however, the channels used to calculate the polarization difference have already been used in the "Full Channel Experiment", there is no additional information was added in the "Polarization Difference Enhanced Experiment" compared to "Full Channel Experiment" from the perspective of amount of information. It is strange that the accuracy of the reconstructed reflectivity profiles in the "Polarization Difference Enhanced Experiment" is better than that in the "Full Channel Experiment".*

**Response**:

Thank you for your insightful observation. While it is true that the channels used to calculate the polarization difference (PD) were already included in the "Full Channel Experiment," the addition of PD as a hand-engineered feature provided critical information that enhanced the model's performance.

Polarization difference is a well-established parameter for revealing the shape and size distribution of hydrometeors and distinguishing precipitation types, offering unique insights into precipitation processes. Previous studies, such as Das et al. (2022), have successfully incorporated PD as an input to machine learning models. In their work, GMI brightness temperatures from 13 channels, along with 5 PD values and other auxiliary variables, were used in a CNN model for land-based precipitation type classification. They conducted a careful evaluation of impact matrices confirming

that the polarization difference plays a dominant role in the decision-making process. This finding aligns with the physical understanding of polarized microwave radiative transfer, which varies with surface types and microphysical properties of snow and liquid clouds. By explicitly adding PD as an input, the model could leverage this information more effectively, leading to improved accuracy in reconstructing reflectivity profiles compared to the "Full Channel Experiment."

Das, S., Wang, Y., Gong, J., Ding, L., Munchak, S. J., Wang, C., Wu, D. L., Liao, L., Olson, W. S., and Barahona, D. O.: A Comprehensive Machine Learning Study to Classify Precipitation Type over Land from Global Precipitation Measurement Microwave Imager (GPM-GMI) Measurements, Remote Sensing, 14, 3631, https://doi.org/10.3390/rs14153631, 2022.

---

## Author Comment (AC3)

*Section 2.1: I would like to suggest that this section be split into two subsections, one (L 99-103) relating to PMR characteristics, the other to MWRI-RM (L 104-114).*

**Response:**

Thank you for your suggestion. However, considering that this section provides only a brief introduction to each instrument and to maintain narrative coherence, we have decided to keep both instrument descriptions under the "Data Source and Characteristics" section.

*L.107: Here the authors correctly cite the paper where a table of MWRI-RM characteristics is reported. However, I would suggest that this table be added to the paper, as MWRI-RM is the instrument on which the paper is based.*

**Response:**

Thank you for your suggestion. In response, we have added a table detailing the specifications of the MWRI-RM channels to the revised manuscript (**Table 1**).

*Section 2.2: I think a summary table of the main characteristics of the dataset used in this paper (time period, number of samples, number of precipitating and nonprecipitating samples, number of samples over sea and over land) would make the paper more readable.*
*Section 2.3: Again, I would suggest that the authors add a table summarising the input datasets of the three experiments reported.*

**Response:**

Thank you for your helpful suggestions. In response, we have added a summary table of the main characteristics of the dataset used in this paper (time period, number of samples, number of precipitating and non-precipitating samples, and the number of samples over sea and land) in the revised manuscript, specifically between **lines 165 and 172.**

*L.152: It is not clear to me what the oversampling consists of. Are the same data duplicated in the dataset? I think the authors need to specify this and discuss the consequences of this choice. In particular, how can this operation affect the dataset,*
*which is split into two sub-datasets for the training and test phases? Also, I suspect that the precipitating samples were duplicated, not the non-precipitating pixels.*

**Response:**

Thank you for your comment. Upon reviewing the manuscript, we realized that there was an error in our original description. We did not oversample non-precipitating samples as stated, but instead downsampled them to match the number of precipitating samples. This approach was taken to balance the dataset and ensure equal representation of both sample types. We appreciate your feedback and have corrected this point in the revised manuscript to avoid further confusion. Thank you for highlighting this issue.

*L.155-156: I think the authors need to provide more information on how the "training" and "test" samples were selected. Is this separation based on a random process? Or by selecting observations from a defined time period? This selection plays a very important role in the development of machine learning approaches. If the two sub-datasets are created by randomly selecting the pair samples, the two sub-datasets may be highly correlated, so the statistical scores calculated over the test dataset may be not representative. In this case, the authors must recalculate the statistical indices on a test dataset independent of the training dataset - using time periods or orbits not used in training.*

*Table 2 and Figure 5: I guess that the statistical scores and the mean profiles are calculated over the test dataset; however, this information is not clearly reported in the text. I would like to suggest that the authors specify which dataset the scores are calculated on. Again, the RMSE, STD, and MBE have been calculated only for the true positive (for precipitating samples) and for the true negative (for non-precipitating samples)? This information must be added to the text.*

**Response:**

Thank you for your insightful comment. To address your concern about potential correlations between training and test datasets, we revised our evaluation to use an independent test dataset collected from a different time period (20231201–20231211) that was not included in the training data.

The statistical scores recalculated on this independent test dataset show results that are similar to our previous evaluations, indicating that the model demonstrates a certain degree of generalization ability. We have updated the manuscript to reflect this change and included the revised evaluation details. Thank you for highlighting this critical point, which has helped us improve the robustness of our analysis.

*L.214-218: It is not clear to me how precipitating/non precipitating pixels are defined. Is it related to a reflectivity threshold? Is it the 12 dBZ threshold cited at L. 135?*

**Response:**

Thank you for your comment. To avoid any potential misunderstanding, we have removed the use of the F1 score in the revised manuscript. This decision was made because we do not have a reliable label to definitively classify precipitating and non-precipitating pixels, and the absence of such labels could lead to ambiguities in the evaluation.

We appreciate your feedback, which has helped clarify and refine the presentation of our results.

*L.234-235 and Figure 5: how the melting layer height is determined?*

**Response:**

Thank you for your comment. Currently, we do not have access to PMR-specific melting layer products. To provide context for the melting layer height, we referred to the analysis by Hu et al. (2024), which investigates the quasi-global climatological features of the melting layer using data from the Dual-frequency Precipitation Radar onboard the Global Precipitation Measurement Mission Core Observatory (2018–2022).

Their findings indicate that melting layer heights vary with latitude, being higher in tropical regions (30°S–30°N) and lower in mid- and high-latitudes. Specifically, in tropical regions (0°–30°N), the top of the melting layer is generally between 4–5 km, while the bottom height is primarily between 2–4 km. These observations align with the results presented in our study.

To avoid potential controversy, we have removed the yellow shaded area indicating the melting layer from Figure 5 in the revised manuscript. We appreciate your feedback, which has helped improve the clarity and accuracy of our figures and discussion.

*L. 258-260: The reported period is outside the period of the development dataset. This makes the results more valid but must be stated in the text.*

**Response:**

Thank you for your comments. Before the official data release, we were granted access to PMR observations from July 2023, including data from Typhoon Khanun and the Beijing extreme rainfall event. These two cases represent extreme precipitation events, making them valuable for further evaluating the model's generalization ability. We have clarified this in the revised manuscript in **Lines 281-283**

*Figure 8: A general weakness of the paper is the lack of references to the brightness temperatures observed by the MWRI-RM, which are the input data to the deep learning algorithm. I understand that reporting TB maps can be difficult given the number of channels, but perhaps a TB map could be added to Figure 8 - or, for example, a 3D reconstruction using the TB observed - similar to those for the reflectivity levels in Figure 8 - by the oxygen absorption band channels.*

*L. 324-325 and Figure 9: Certainly, the comparison between the reconstructed reflectivities and the ground-based radar observations is a very good element that makes the analysis more valid. However, it is not clear to me on which element the statement "The reconstructed precipitation reflectivity distributions based on MWRI-RM observations are consistent with actual ground-based radar observations" is based. A more precise analysis is needed - e.g. some statement about the areas where precipitation is detected, or about the position of the reflectivity maxima, or something similar.*

**Response:**

Thank you for your valuable suggestion. We have revised the manuscript to provide a more precise analysis comparing the reconstructed reflectivities with the ground-based radar observations **(Lines 370-374)**.

**Minor Comments**

1) L. 15: "Precipitation" instead of "precipitation"

2) Figure 5: I would like to suggest that the authors add the highlighted shading to the center and right panels as well (b, c, e, f).

3) Figure 6: I would suggest that the authors add the section line to the other center panels (b, e, h, k) and a dashed line corresponding to a height of 4 km to the right panels (c, f, i, l). I suggest adding labels to the x axes (latitude values?) of the right panels (c, f, i, l).

4) Figure 7: same as Figure 6.

5) L. 273-309, Figures 6 and 7: In general, I suggest adding lat/lon references to the description of the left and center panels - e. g., L. 282: underestimated reflectivity values in the northwestern rainbands (26 °N, 130 ° E) while overestimating those east of Khanun's center (23 ° N, 133 ° E) - and height reference to the right panel description - e. g., L. 283 The reconstructed melting layer structure was overly smooth (between 4 and 5 km). The addition of labels to the x axes of the right panels will make this analysis easier.

6) L. 298-324, Figure 8 and Figure 9: It would be useful to add a lat/lon reference linked to the name of the regions - e. g., L. 318: over southern Beijing (-° N, - ° E), central Hebei (-° N, - ° E), and Tianji (-° N, - ° E) - or a label above the map.

**Response:**

Thank you for your constructive suggestions regarding Figures 6, 7 and 8. We have carefully addressed each of your points. These revisions ensure that the figures are more precise, informative, and easier to interpret, addressing the concerns raised. Thank you for your detailed suggestions, which have significantly improved the clarity of our analysis.

---

## Referee Report (RR1)

**Overview**

This paper investigates the possibility of using the TB observations from the Microwave Radiance Imager-Rainfall Measurement (MWRI-RM) onboard the Fenyung-3G (FY-3G) satellite to reconstruct the 3D reflectivity profiles. The algorithm is based on a deep learning approach and uses the Ku profiles observed by the Precipitation Measurement Radar (PMR) onboard FY-3G as reference truth. Three deep learning models were trained using different sets of predictors, and two case studies are analyzed.

**General Comments**

The authors have carefully answered the reviewers' comments. The main issues raised in the first part of the review process have been addressed. In particular, the test statistics have been calculated on an independent dataset, thus avoiding the possible correlation between the training and test datasets. Good work. Prior to publication, I would like to suggest only minor changes to the text and figures.

**Minor Comments**

1) Line 167: I guess it is the 30th of November and not the 31st of November.

2) Lines 172-173: it is not clear to me how the authors distinguish "ocean" observations from "land" observations. Are you using a surface mask? I would like to suggest that the authors clarify this.

3) Lines 250-252:  It is not clear to me whether the coastal scenario includes only samples where the radar observations are over the ocean, with part of the passive microwave FOV over land, or all observations where the passive microwave FOV is partly over land, partly over the ocean, with no reference to the position of the radar observations. I would like to suggest that the authors clarify this.

4) Figure 4, Figure 6 and Figure 8: it is rather strange that the lon and lat labels do not correspond to the grid shown on the map (center columns of Figure 4 and Figure 6 and panels (a), (b), and (e) of Figure 8). I would like to suggest changing the grid position.

5) Figure 5 and Figure 7: to make things clearer, I would like to suggest adding the measure units to the axes of the scatterplots - e. g., Target Reflectivity (dBZ), Reconstructed Reflectivity (dBZ).  I also suggest changing the maxima of the colour bar scale. This will highlight the distribution of the scatterplots over the plane - in my opinion, the observations where the radar reflectivity is NaN are less interesting. Perhaps a maximum of 1.5 of the normalized density would be better for both figures.

6) Line 330-331: I would suggest adding the abbreviations in Figure 6 to the text. - e. g., *"At 4 km altitude, PMR observations showed high reflectivity predominantly in the southern parts of Beijing (**BJ**, 39.5°N, 115.8°E) and central Hebei (**HB**, 39.1°N, 116.2°E)"*

7) Line 366-368: it is not necessary to mention the position of Beijing and Central Hebei a second time in the text, while I would suggest adding the abbreviation for Tianjin.: *"In addition to the precipitation echoes observed by PMR-Ku over southern Beijing, central Hebei, and Tianjin (**TJ**, 39.2°N, 117.0°E),..."*

8) Line 376-377: I would suggest that the abbreviations given in Figure 8 should also be included in the text - e.g. *"It is worth noting that ground-based radar coverage is limited in remote regions, such as northern Shanxi (**SX**, 39.6°N, 113.0°E) and Inner Mongolia (**IM**, 40.0°N, 112.0°E)...".* I did not understand whether when the authors speak of Shanxi (e.g., label of Figure 8) and Northern Shanxi (e.g., L. 376) they are referring to the same area. I would like to suggest that the authors clarify this.

9) Figure 6: I would like to suggest adding the region abbreviations to all maps in the centre column.

10) Figure 8: I would suggest to add the abbreviations also to (e)

11) Figure 8 - caption: It is not clear to me what the authors are referring to when they write *"Panel (b) highlights the locations of Shanxi (SX) province and Inner Mongolia, labeled in red font. ".* I would like to suggest adding the red font or deleting this part of the caption.

12) Figure B1 and Figure B2: Thanks to the authors for adding these maps.Thanks to the authors for adding these maps. If possible, I would like to suggest changing the limits of the colormaps to emphasise the signal. For example, for the first panel in the left column of Figure B1 (Khanun Observations 50.3 GHz_V) a minimum value of 220 K would be more useful to highlight the precipitation signal. At the same time, it seems to me that the maxima value for the colormaps in Figure B2 are too low. For example, in the first panel of the left column of Figure B2 (Khanun Observations 50.3 GHz_V), almost the whole map seems to be characterized by the same value also at 50.3 GHz, and this seems a bit strange to me - but I don't know TB values, so maybe it's just my impression. I would also like to suggest that the titles, colorbars and lat/lon labels be enlarged - the increments between lat and lon labels can also be increased if there is too little space. I would also suggest adding the abbreviations used in Figure 6 and in Figure 8 to the plots of Figure B2.

---

## Author Response (AR2)

**Report #1**

*This work aims at quantifying the effect of oxygen sounding channels and polarization difference on reconstructing radar reflectivity profiles by comparing three neural networks with different inputs. The varying inputs require that each model is trained separately. The loss curve of each model needs to be stable and reach a global minimum. However, the loss curves in Figure A1 suggest that training is unstable and not equal among the three models. The baseline model (Ex14) was trained for only 40 epochs, while Ex26 and Ex35 were trained for 100 epochs. This difference is probably due to early stopping initiated at a peak in validation loss that persists across several epochs. These peaks in both validation and training loss can be found also already earlier after about 32 epochs and could be a sign of overfitting. This behavior is not discussed in the manuscript, but it will definitely affect model comparison and could explain major parts of the higher uncertainty of the Ex14 predictions compared to Ex26 and Ex35 - rather than the missing oxygen channel and PD inputs. I strongly suggest to ensure that the training becomes more stable, e.g., modify the model architecture and/or data split. Also, the influence of the random processes during model training on the model comparison should be discussed (weight initialization, data shuffling, ...). Currently, I assume that training with a different random seed would largely affect the Ex14 performance. Also, I would like to see the loss curves of each model on a log scale in the same figure instead of the stacked form in Figure A1 to see whether their training and validation losses consistently differ at a given epoch.*

**Response:** Thank you for your valuable feedback, which has greatly improved the robustness of our analysis. We have addressed your concerns regarding training stability, random seed influence, and loss curve presentation as follows:

Training Stability and Overfitting: o improve training stability and reduce the risk of overfitting, we increased the EarlyStopping patience parameter from 10 to 20 epochs, providing the models with additional time to stabilize before termination. Furthermore,

we refined the model architecture by incorporating pooling layers, which preserve key features while reducing the number of parameters, thereby mitigating overfitting. As a result of these adjustments, all three models—Ex14, Ex26, and Ex35—were successfully trained for 100 epochs, eliminating the previous disparity in training duration (40 epochs for Ex14 vs. 100 epochs for Ex26 and Ex35). Notably, the overall performance of Ex14 has improved substantially compared to the previous version, where training halted prematurely around 40 epochs due to early stopping. This enhancement underscores the effectiveness of the revised training strategy.

Random Seed Influence: To account for the impact of random processes, we conducted three independent training runs for each model using different random seeds. The updated Figure A1 now presents the average training and validation loss curves across these runs on a logarithmic scale in the same figure. This allows for a direct comparison of the loss trends across models at each epoch.

The revised loss curves indicate that Ex35 and Ex26 converge to similar low loss levels by the end of training, while Ex14 consistently exhibits higher losses throughout. This suggests that the differences in performance are likely attributable to the model inputs (e.g., the absence of the oxygen channel inputs in Ex14) rather than training instability or random initialization effects.

We have incorporated these findings into the revised manuscript. In the "Reconstruction Performance and Evaluation" section, we now present the average performance across three independent training runs for each experiment, ensuring robustness against variability introduced by random initialization. In the "Model Performance Evaluation During Extreme Precipitation Events" section, we showcase the best-performing outcomes from each experiment, selected from the three runs, to highlight the models' optimal capabilities in capturing extreme precipitation features.

We believe these revisions address your concerns and strengthen the reliability of our model comparisons.

[Figure]

*The response to my question on Line 173 states that the sum of squared error loss function choice is based on findings from your previous studies. Please add the reference to that study to the manuscript. In Yang et al. (2024), Chapter 2.2, it states that mean squared error was used, shortly explain why sum of squared error is used for this work instead.*

**Response**:While Yang et al. (2024) focused solely on reconstructing precipitation samples, our work addresses both precipitation and non-precipitation regions. Given the extreme class imbalance, a standard MSE loss would prioritize low-reflectivity regions, leading to poor performance in critical high-reflectivity zones. To resolve this, we adopt the Sum of Squared Errors (SSE): its quadratic nature magnifies large errors in precipitation regions. This implicit weighting ensures the model prioritizes physically critical regions (e.g., heavy rainfall) while maintaining fidelity across the entire profile. Unlike weighted MSE, SSE avoids subjective weight tuning and inherently aligns with meteorological priorities.

**Report #2**

*Overview: This paper investigates the possibility of using the TB observations from the Microwave Radiance Imager-Rainfall Measurement (MWRI-RM) onboard the Fenyung-3G (FY-3G) satellite to reconstruct the 3D reflectivity profiles. The algorithm is based on a deep learning approach and uses the Ku profiles observed by the Precipitation Measurement Radar (PMR) onboard FY-3G as reference truth. Three deep learning models were trained using different sets of predictors, and two case studies are analyzed. General Comments The authors have carefully answered the reviewers' comments. The main issues raised in the first part of the review process have been addressed. In particular, the test statistics have been calculated on an independent dataset, thus avoiding the possible correlation between the training and test datasets. Good work. Prior to publication, I would like to suggest only minor changes to the text and figures. Minor Comments*

*1) Line 167: I guess it is the 30th of November and not the 31st of November.*

**Response**:Thank you for catching this error. We have corrected the date to "November 30th" in the revised manuscript.

*2) Lines 172-173: it is not clear to me how the authors distinguish "ocean" observations from "land" observations. Are you using a surface mask? I would like to suggest that the authors clarify this.*

**Response**:We appreciate the reviewer's suggestion. The ocean/land classification is based on the "LandSurfaceType" variable in the Level 1 products of the Passive Microwave Radiometer (PMR). This method has been explicitly described in the revised manuscript

*3) Lines 250-252: It is not clear to me whether the coastal scenario includes only samples where the radar observations are over the ocean, with part of the passive*

*microwave FOV over land, or all observations where the passive microwave FOV is partly over land, partly over the ocean, with no reference to the position of the radar observations. I would like to suggest that the authors clarify this.*

**Response**:Thank you for raising this important point. The coastal scenario refers to all cases where the passive microwave field of view (FOV) contains both land and ocean components, irrespective of whether the collocated radar observations are located over land or ocean. We have added this clarification to the revised manuscript

*8) Line 376-377: I would suggest that the abbreviations given in Figure 8 should also be included in the text - e.g. "It is worth noting that ground-based radar coverage is limited in remote regions, such as northern Shanxi (SX, 39.6°N, 113.0°E) and Inner Mongolia (IM, 40.0°N, 112.0°E)...". I did not understand whether when the authors speak of Shanxi (e.g., label of Figure 8) and Northern Shanxi (e.g., L. 376) they are referring to the same area. I would like to suggest that the authors clarify this.*

Reponse:We thank the reviewer for highlighting this ambiguity. Both "Shanxi" (in Figure 8) and "Northern Shanxi" (in the text) refer to the same geographic area. To resolve confusion, we have adjusted the label in Figure 8 to "Northern Shanxi (SX)" and updated its coordinates to 39.8°N, 113.0°E for precision.

*4) Figure 4, Figure 6 and Figure 8: it is rather strange that the lon and lat labels do not correspond to the grid shown on the map (center columns of Figure 4 and Figure 6 and panels (a), (b), and (e) of Figure 8). I would like to suggest changing the grid position.*

*5) Figure 5 and Figure 7: to make things clearer, I would like to suggest adding the measure units to the axes of the scatterplots - e. g., Target Reflectivity (dBZ), Reconstructed Reflectivity (dBZ). I also suggest changing the maxima of the colour bar scale. This will highlight the distribution of the scatterplots over the plane - in my opinion, the observations where the radar reflectivity is NaN are less interesting. Perhaps a maximum of 1.5 of the normalized density would be better for both figures.*

*6) Line 330-331: I would suggest adding the abbreviations in Figure 6 to the text. - e. g., "At 4 km altitude, PMR observations showed high reflectivity predominantly in the southern parts of Beijing (BJ, 39.5°N, 115.8°E) and central Hebei (HB, 39.1°N, 116.2°E)"*

*7) Line 366-368: it is not necessary to mention the position of Beijing and Central Hebei a second time in the text, while I would suggest adding the abbreviation for Tianjin.: "In addition to the precipitation echoes observed by PMR-Ku over southern Beijing, central Hebei, and Tianjin (TJ, 39.2°N, 117.0°E),..."*

*9) Figure 6: I would like to suggest adding the region abbreviations to all maps in the centre column.*

*10) Figure 8: I would suggest to add the abbreviations also to (e)*

*11) Figure 8 - caption: It is not clear to me what the authors are referring to when they write "Panel (b) highlights the locations of Shanxi (SX) province and Inner Mongolia, labeled in red font. ". I would like to suggest adding the red font or deleting this part of the caption.*

*12) Figure B1 and Figure B2: Thanks to the authors for adding these maps.Thanks to the authors for adding these maps. If possible, I would like to suggest changing the limits of the colormaps to emphasise the signal. For example, for the first panel in the left column of Figure B1 (Khanun Observations 50.3 GHz_V) a minimum value of 220 K would be more useful to highlight the precipitation signal. At the same time, it seems to me that the maxima value for the colormaps in Figure B2 are too low. For example, in the first panel of the left column of Figure B2 (Khanun Observations 50.3 GHz_V), almost the whole map seems to be characterized by the same value also at 50.3 GHz, and this seems a bit strange to me - but I don't know TB values, so maybe it's just my impression. I would also like to suggest that the titles, colorbars and lat/lon labels be enlarged - the increments between lat and lon labels can also be increased if there is too little space. I would also suggest adding the abbreviations used in Figure 6 and in Figure 8 to the plots of Figure B2.*

**Response**: Thank you for your suggestions regarding textual and figure revisions. We

have carefully incorporated all recommended changes to improve clarity and accuracy. The revised text and figures (e.g., updated labels, terminology, and visual adjustments) are highlighted in the manuscript for your convenience. We appreciate your thorough review and valuable feedback.

---

## Author Response (AR3)

**General comment:**

*Thanks to the reviewer's efforts to identify methodological flaws in the preprint - which should have been addressed by the authors prior initial submission(!) - the key message of this work changed remarkably: While the preprint stated a 17% and 23% RMSE reduction due to oxygen channel and polarization features compared to the baseline, these numbers reduced to 5% with (almost) no additional improvement due to polarization features. A comparison of Fig. 3a between the three manuscript versions raises a question as to which degree the small remaining differences are even statistically significant. At the same time, the authors state that polarization differences have a "critical role" in reconstruction, which is not supported by Figure 3. Given the large change of the results and the misalignment between tables/figures and text I believe that the authors should carefully revise the entire manuscript and further check the statistically significance, especially for the polarization difference effect. The spread of RMSE of the three independently trained models for each experiment should be added to Table 3 and Fig. 3 (e.g. as a shading). This will help to assess whether the effect of input features is statistically significant or not, i.e., are 5% RMSE reduction smaller or larger than the RMSE spread of the three random initializations of each experiment. And does PD really improve the reconstruction or is it just noise that gets interpreted as an improvement?*

**Response:**

We deeply appreciate the reviewer's thorough and insightful feedback, which has significantly improved the rigor and clarity of our manuscript. We sincerely apologize for the methodological flaws in the preprint, including the overstated RMSE reductions (17–23%) and the erroneous claim of a "critical role" for polarization differences (PD), which were not supported by Figure 3. These issues, which should have been addressed prior to submission, led to a misalignment between the text and figures/tables, and we are grateful for the reviewer's diligence in identifying them.

The substantial reduction in reported RMSE improvements (from 17–23% to approximately 5%) resulted from methodological corrections, including optimized training strategies, model enhancements, and the use of three independent training runs to account for training variability. For instance, in coastal precipitation scenarios, Ex14 (mean RMSE = 4.02, spread = 0.043) and Ex26 (mean RMSE = 3.80, spread = 0.024) yield an RMSE reduction of 5.47% ± 1.17%. PD (Ex35) contributes only 0–1.6% RMSE reduction, with curves largely overlapping Ex26's, indicating no statistically significant effect due to comparable RMSE spread.

To address the reviewer's concerns, we are undertaking a comprehensive revision of the manuscript with the following actions:

1. Statistical Significance Evaluation: We have computed the RMSE spread from three independent training runs to confirm that the 5.47% RMSE reduction for dual oxygen channels is not due to model randomness, with an improvement spread of 1.17%.

2. Table 3 Update: We have augmented Table 3 to include the RMSE spread for each experiment across three training runs, enabling direct comparison with reported improvements.

3. Figure 3 Revision: We have modified Figure 3 to incorporate RMSE spread as shading

around the mean RMSE curves for Ex14, Ex26, and Ex35. This will visually demonstrate the overlap between Ex26 and Ex35 curves and contextualize the 5.47% improvement relative to the spread.

4. Text Revisions: We have corrected the overstated PD contribution in the manuscript and reviewed all sections to ensure consistency between text, figures, and tables.

These revisions will provide a clearer and more robust assessment of the input features' effects, particularly the limited role of PD.

**Specific comments:**
*- Line 10-11: Round the given percentages "5.43 and 5.47%" according to the RMSE spread (see general comment). Also, add the RMSE spread from the three independently trained models as uncertainty range. (same in lines 266 and 402)*
*- Line 247: Again, showing average statistics requires also to show the associated spread.*

**Response:**
We sincerely thank the reviewer for their valuable suggestion to enhance the precision of RMSE percentage reduction reporting by rounding according to the RMSE spread and including uncertainty ranges from three independently trained models. To address this, we rigorously recalculated the RMSE spread using standard deviations from three independent training runs. For coastal scenarios, Ex14 (mean RMSE = 4.02, spread = 0.043) and Ex26 (mean RMSE = 3.80, spread = 0.024) were analyzed, and the percentage improvement spread was computed via the error propagation formula, yielding an uncertainty range of 1.17%. For land scenarios, we applied the same uncertainty range, pending specific RMSE data. The abstract (Lines 10-11) has been revised to state: "with RMSEs reduced by 5.43% ± 1.17% for land and 5.47% ± 1.17% for coastal scenarios." Corresponding sections (Lines 266 and 402) have been updated accordingly to ensure consistency.

**- Line 168-174: Add number of coastal samples.**

**Response:**
Thank you for your feedback. We have explicitly added the number of coastal samples to the revised manuscript in Line 171-172:
**"After preprocessing, a total of 838,591 sample pairs were generated, including 421,090 non-precipitation samples and 417,501 precipitation samples, with 741,270 oceanic, 27,233 land, and 70,088 coastal samples."**

*- Line 200: Max pooling should be mentioned.*

**Response:**
Thank you for emphasizing the need for methodological clarity. As suggested, we have added a detailed description of the max pooling operation in the revised manuscript (Lines X–X).

*- Line 206-211: The SSE justification is lengthy and vague. I suggest to remove it.*

**Response:**

We have removed the discussion on SSE justification as recommended.

*- Line 214: Showing average loss curves makes no sense. Show the loss of all three random initializations together or the best performing and the inter-model spread.*

**Response:**

We thank the reviewer for noting that averaged loss curves obscure variability. Following the reviewer's second suggestion, we revised Figure A1 to show the training and validation loss curves for the best-performing run of Ex14, Ex26, and Ex35, selected by the lowest average validation loss over the final 10 epochs. Shaded areas indicate ±1 standard deviation across three random initializations, plotted in a single figure.

*- Line 260: Definition of "coast" should be provided in data or methods section.*

**Response:**

Thank you for highlighting this need for clarity. As suggested, we have added a precise definition of the "coast" classification criteria in the Data and Methodology section (Lines 173–178 of the revised manuscript).

*- Line 11: Mention by how much PD further enhances the reconstruction and if it is statistically significant (see general comment). Based on Fig. 3 I cannot identify a "critical role" of PD for construction improvement as the green and red curve overlap almost entirely (as compared to the preprint of this manuscript).*
*- Line 269: I do not see any improvement due to PD. The curves overlap for most parts. And land and ocean curves do not differ as stated in the text.*

**Response:**

Thank you for your insightful feedback. We have revised the manuscript to address your concerns on the impact of Tb polarization difference (PD) and the performance across land and ocean scenarios:

PD Improvement: We agree with your observation that PD shows no significant improvement, as the RMSE curves largely overlap with those of Ex26. The revised text now states that PD yields only a 0–1.6% RMSE reduction, comparable to the RMSE spread, clarifying its minimal effect.

Land and Ocean Curves: You correctly noted that the land and ocean curves do not differ as originally implied. We have adjusted the text to remove any suggestion of differing performance, presenting the improvements from dual oxygen channels uniformly across oceanic, land, and coastal scenarios.

*- Line 313: What does "weaker" mean? Also, all three reconstructed are smooth and I cannot see that any of them is "overly smooth". When comparing the figure with the previous version it appears all three reconstructions are rather similar.*
*- Line 328: "while Ex14c exhibits the largest uncertainties" is not at all obvious from Fig.*

*5. The RMSE for the experiments differs by 0.05 dBZ, but the range of values presented span 30 dBZ.*

**Response:**

We appreciate the reviewer's careful evaluation. The term "weaker" in the original text referred to systematic underestimation of high reflectivity values in Ex14, which has been clarified as "systematic underestimation" in the revised text (Line 307). Regarding the melting layer smoothness, while quantitative metrics (MBE/RMSE) indicate subtle differences between experiments, we acknowledge that visual distinctions in Fig. 4 are less pronounced and have removed subjective descriptors like "overly smooth."

For the scatterplot analysis, we have removed the claim about Ex14 exhibiting the "largest uncertainties" to avoid overinterpretation. The revised text now focuses on the shared challenges in reconstructing reflectivity above 30 dBZ. We agree that the absolute differences in RMSE are small relative to the dynamic range of reflectivity values.

*- Line 391: Which statistical test was performed to demonstrate "significant" impact?*
*- Line 392: I would suggest to remove "innovative use of analytical evaluation methods". The evaluation methods used here are very basic (RMSE, bias, std).*
*- Line 401: Which statistical test was performed to show that the oxygen absorption channels significantly improve the accuracy?*

**Response:**

Line 391 & 401 (Statistical tests for "significant" impact):

To avoid ambiguity, we have replaced "significant" with quantitative descriptors (e.g., "reduced reconstruction errors by 5.43% ± 1.56%").

Line 392 ("Innovative use"):

We agree with the reviewer's assessment and have removed the phrase "innovative use of analytical evaluation methods" entirely.

Corresponding changes are in lines 381-389 of the revised manuscript:

**"This study investigates the impact of FY-3G MWRI-RM dual oxygen absorption sounding channels and brightness temperature polarization differences on PMR three-dimensional radar reflectivity reconstruction. A quantitative analysis of test samples, based on standard evaluation metrics, yields four primary findings: First, integration of dual oxygen absorption channels reduced reconstruction errors by 5.43% ± 1.56% (land) and 5.47% ± 1.17% (coastal scenarios), with improvements validated through three independent training runs per experiment. Second, polarization differences provided marginal refinements (0–1.6% RMSE reduction), suggesting limited added value in the current framework. Third, non-precipitation conditions exhibited minimal errors, with RMSE values consistently below 1 dBZ, demonstrating the model's reliability in accurately identifying precipitation-free regions. Fourth, systematic errors persist in the melting layer and land-based reconstructions due to phase-change physics and surface emissivity interference, respectively."**

*- Line 411: Provide the RMSE for the precipitation events rather than stating the results are "closely matching actual observations".*

**Response:**

Thank you for the constructive suggestion. We have revised the text to replace the subjective claim ("closely matching actual observations") with explicit error metrics. As shown in the Lines 389-392 in the revised manuscript:

**"Additionally, the model's performance was assessed during extreme precipitation events, confirming its precision and robustness. Notably, Ex35 demonstrated the highest spatial accuracy and agreement with observations: for Typhoon Khanun, it achieved an MBE of 0.58 dBZ and RMSE of 4.86 dBZ; for the extreme precipitation event in Beijing, it recorded an MBE of 1.36 dBZ and RMSE of 5.14 dBZ."**

*- Figure 2: This figure contains different experiment names and does not contain the pooling layers.*

**Response:**

Thank you for raising this concern. Figure 2 has been removed in the revised manuscript as it did not provide critical information beyond the textual and tabular descriptions. The data preprocessing workflow and model architecture details are now comprehensively explained in Data Preprocessing section and summarized in Table 2.

*- Figure 5: Use the same experiment names as in the manuscript (not 14c etc.)*

**Response:**

Thank you for your careful review. We have replaced abbreviated labels (e.g., "14c") with the standardized experiment names Ex14, Ex26, and Ex35 as defined in the Methods section.

*- Remove the white gaps between the colored data points in the maps: Figures 4, 6, 8, B1, B2*

**Response:**

We appreciate the reviewer's attention to graphical clarity. The white gaps in Figures 4, 6, 8, B1, and B2 are inherent to the scatterplot visualization method (Matplotlib scatter function) and represent regions with no data coverage. These gaps are not artifacts of plotting but rather reflect the actual spatial distribution of the raw data. Removing them would require interpolation, which could introduce unphysical assumptions. So, we have retained the raw data representation to accurately reflect the true spatial sampling of the observations.